# A Solvable High-Dimensional Model of GAN

**Chuang Wang**[1,2]
wangchuang@ia.ac.cn

**Hong Hu**[2]
honghu@g.harvard.edu

**Yue M. Lu**[2]
yuelu@seas.harvard.edu

1. National Laboratory of Pattern Recognition, Institute of Automation,
Chinese Academy of Science, 95 Zhong Guan Cun Dong Lu, Beijing 100190, China
2. John A. Paulson School of Engineering and Applied Sciences, Harvard University
33 Oxford Street, Cambridge, MA 02138, USA

## Abstract

We present a theoretical analysis of the training process for a single-layer GAN fed by high-dimensional input data. The training dynamics of the proposed model at both microscopic and macroscopic scales can be exactly analyzed in the high-dimensional limit. In particular, we prove that the macroscopic quantities measuring the quality of the training process converge to a deterministic process characterized by an ordinary differential equation (ODE), whereas the microscopic states containing all the detailed weights remain stochastic, whose dynamics can be described by a stochastic differential equation (SDE). This analysis provides a new perspective different from recent analyses in the limit of small learning rate, where the microscopic state is always considered deterministic, and the contribution of noise is ignored. From our analysis, we show that the level of the background noise is essential to the convergence of the training process: setting the noise level too strong leads to failure of feature recovery, whereas setting the noise too weak causes oscillation. Although this work focuses on a simple copy model of GAN, we believe the analysis methods and insights developed here would prove useful in the theoretical understanding of other variants of GANs with more advanced training algorithms.

## 1 Introduction

A generative adversarial network (GAN) [1] seeks to learn a high-dimensional probability distribution from samples. While there have been numerous advances on the application front [2–6], considerably less is known about the underlying theory and conditions that can explain or guarantee the successful trainings of GANs.

Recently, it has been a very active area of research to study either the equilibrium properties [7–9] or the training dynamics [10, 11]. Specifically, there is a line of works studying the dynamics of the gradient-based training algorithms *e.g.*, [11–16]. The basic idea is the following. The evolution of the learnable parameters in the training dynamics can be considered as a discrete-time process. With a proper time scaling, this discrete-time process converges to a deterministic continuous-time process as the learning rates tend to 0, which is characterized by an ordinary differential equation (ODE). By studying local stability of the ODE's fixed points, [12] shows that oscillation in the training algorithm is due to the eigenvalues of the Jacobian of the gradient vector field with zero real part and large imaginary part. Due to this fact, various stabilization approaches are proposed, for example adding additional regularizers [13, 14], and using two timescale [15] training. Very recently, [16] argues that those stabilization techniques may encourage the algorithms to converge non-Nash stationary points. All above works consider a small-learning-rates limit, where the limiting process

is always deterministic. The stochasticity and the effect of the noise is essentially ignored, which may not reflect practical situations. Thus, a new analysis paradigm to study the dynamics with the consideration of the intrinsic stochasticity is needed.

In this paper, we present a *high-dimensional* and *exactly solvable* model of GAN. Its dynamics can be precisely characterized at both macroscopic and microscopic scales, where the former is deterministic and the latter remains stochastic. Interestingly, our theoretical analysis shows that injecting additional noise can stabilize the training. Specifically, our main technical contributions are twofold:

- We present an asymptotically exact analysis of the training process of the proposed GAN model. Our analysis is carried out on both the *macroscopic* and the *microscopic* levels. The macroscopic state measures the overall performance of the training process, whereas the microscopic state contains all the detailed weights information. In the high-dimensional limit ($n \to \infty$), we show that the former converges to a deterministic process governed by an ordinary differential equation (ODE), whereas the latter stays stochastic described by a stochastic differential equation (SDE).

- We show that depending on the choice of the learning rates and the strength of noise, the training process can reach either a successful, a failed, an oscillating, or a mode-collapsing phase. By studying the stabilities of the fixed points of the limiting ODEs, we precisely characterize when each phase takes place. The analysis reveals a condition on the learning rates and the noise strength for successful training. We show that the level of the background noise is essential to the convergence of the training process: setting the noise level too strong (small signal-to-noise ratio) leads to failure of feature recovery, whereas setting the noise too weak (large signal-to-noise ratio) causes oscillation.

Our work builds upon a general analysis framework [17] for studying the scaling limits of high-dimensional exchangeable stochastic processes with applications to nonlinear regression problems. Similar techniques have also been used in the literature to study Monte Carlo methods [18], online perceptron learning [19, 20], online sparse PCA [21], subspace estimation [22], online ICA [23] and more recently, the supervised learning of two-layer neural networks [24], but to our best knowledge, this technique has not yet been used in analyzing GANs.

The rest of the paper is organized as follows. We present the proposed GAN model and the associated training algorithm in Section 2. Our main results are presented in Section 3, where we show that the macroscopic and microscopic dynamics of the training process converge to their respective limiting processes that are characterized by an ODE and SDE, respectively. In Section 4, we analyze the stationary solutions of the limiting ODEs and precisely characterizes the long-term behaviors of the training process. We conclude in Section 5.

## 2  Formulations

In this section, we introduce the proposed GAN model and specify the associated training algorithm.

**Model for the real data.**  In order to establish the theoretical analysis, we first impose a model for the probability distribution from which we draw our real data samples. We assume that the real data $\boldsymbol{y}_k \in \mathbb{R}^n$, $k = 0, 1, \ldots$ are drawn according to the following generative model:

$$\boldsymbol{y}_k = \mathcal{G}(\boldsymbol{c}_k, \boldsymbol{a}_k; \boldsymbol{U}, \eta_{\mathrm{T}}) \stackrel{\text{def}}{=} \boldsymbol{U}\boldsymbol{c}_k + \sqrt{\eta_{\mathrm{T}}}\boldsymbol{a}_k, \tag{1}$$

where $\boldsymbol{U} \in \mathbb{R}^{n \times d}$ is a deterministic unknown feature matrix with $d$ features; $\boldsymbol{c}_k \in \mathbb{R}^d$ is a random vector drawn from an unknown distribution $\mathcal{P}_{\boldsymbol{c}}$; $\boldsymbol{a}_k$ is an $n$-dimensional random vector acting as the background noise; and $\eta_{\mathrm{T}}$ is a parameter to control the strength of noise. Without loss of generality [1], we assume $\boldsymbol{U}^\top \boldsymbol{U} = \boldsymbol{I}_d$, where $\boldsymbol{I}_d$ is the $d \times d$ identity matrix.

This generative model, referred to as the spiked covariance model [25] in the literature, is commonly used in the theoretical study of principal component analysis (PCA). We note that this model is not a trivial task for PCA even when $d = 1$ if the variance of the noise $\boldsymbol{a}_k$ is a non-zero constant. As

proved in [25], the best estimator can not perfectly recover the signal $U$ given an $\mathcal{O}(n)$ number of samples $y_k$. Thus, it is of sufficient interest to investigate whether a GAN can retrieve informative results for the principal components in the same scaling limit.

**The GAN model**    The GAN we are going to analyze is defined as follows. We assume that the generator $\mathcal{G}$ has the same linear structure as the real data model (1) given above:

$$\widetilde{y}_k = \mathcal{G}(\widetilde{c}_k, \widetilde{a}_k; V, \eta_{\mathrm{G}}) \qquad (2)$$

but the parameters are different. Here, $\widetilde{y}_k$ denotes a fake sample produced by the generator; $\widetilde{a}_k$ is an $n$-dimensional random noise vector; the random variable $\widetilde{c}_k$ is drawn from a fixed distribution $\mathcal{P}_{\widetilde{c}}$; $\eta_{\mathrm{G}}$ is the noise strength; and the matrix $V \in \mathbb{R}^{n \times d}$ represents the parameters of the generator. (In an ideal case in which the generator learns the underlying true probability distribution perfectly, we have $V = U$.) Throughout the paper, we follow the notational convention that all the symbols that are decorated with a tilde (*e.g.*, $\widetilde{y}_k, \widetilde{c}_k, \widetilde{a}_k$) denote quantities associated with the generator.

We define the discriminator $\mathcal{D}$ of our GAN model as

$$\mathcal{D}(y; w) \overset{\text{def}}{=} \widehat{D}(y^\top w).$$

Here, $y$ is an input vector, which can be either the real data $y_k$ from (1) or the fake one $\widetilde{y}_k$ from (2); $\widehat{D} : \mathbb{R} \mapsto \mathbb{R}$ can be any function; and the vector $w \in \mathbb{R}^n$ represents the parameters associated with the discriminator. Later, we will show that the generator can learn multiple features even though the discriminator only has one feature vector $w$. Discriminators with multiple features can also be analyzed in a similar way, but in this paper we consider the single-feature discriminator for simplicity.

**The training algorithm.**    The proposed GAN model has two set of parameters $V$ and $w$ to be learned from the data. The training process is formulated as the following MinMax problem

$$\min_{V} \max_{w} \mathbb{E}_{y \sim \mathcal{P}(y;U)} \mathbb{E}_{\widetilde{y} \sim \widetilde{\mathcal{P}}(\widetilde{y},V)} \mathcal{L}(y, \widetilde{y}; w), \qquad (3)$$

where the two probability distributions $\mathcal{P}(y; U)$ and $\widetilde{\mathcal{P}}(\widetilde{y}; V)$ represent the distributions of the real data $y$ and the fake data $\widetilde{y}$ as specified by (1) and (2) respectively, and

$$\mathcal{L}(y, \widetilde{y}; w) \overset{\text{def}}{=} F(\widehat{D}(y^\top w)) - \widetilde{F}(\widehat{D}(\widetilde{y}^\top w)) - \tfrac{\lambda}{2} H(w^\top w) + \tfrac{\lambda}{2} \mathrm{tr}\big(H(V^\top V)\big) \qquad (4)$$

with $F(\cdot)$ and $\widetilde{F}(\cdot)$ being two functions that quantify the performance of the discriminator and $\lambda > 0$ being a constant. The function $H(\cdot)$ acts as a regularization term introduced to control the magnitude of the parameters $w$ and $V$. It can be an arbitrary real-valued function, which is applied element-wisely if the input is a matrix.

We consider a standard training algorithm that uses the vanilla stochastic gradient descent/ascent (SGDA) to seek a solution of (3). To simplify the theoretical analysis, we consider an online (*i.e.*, streaming) setting where each data sample $y_k$ is used only once. At step $k$, the model parameters $w_k$ and $V_k$ are updated using a new real sample $y_k$ and two fake samples $\widetilde{y}_{2k}$ and $\widetilde{y}_{2k+1}$, according to

$$
\begin{aligned}
w_{k+1} &= w_k + \tfrac{\tau}{n} \nabla_{w_k} \mathcal{L}(y_k, \widetilde{y}_{2k}; w_k) \\
V_{k+1} &= V_k - \tfrac{\widetilde{\tau}}{n} \nabla_{V_k} \mathcal{L}\big(y_k, \mathcal{G}(\widetilde{c}_{2k+1}, \widetilde{a}_{2k+1}; V_k; \eta_{\mathrm{G}}); w_k\big),
\end{aligned}
\qquad (5)
$$

where $\widetilde{c}_{2k+1}, \widetilde{a}_{2k+1}$ are random variables that generates the fake sample $\widetilde{y}_{2k+1}$ according to (2). The two parameters $\tau$ and $\widetilde{\tau}$ in the above expressions control the learning rates of the discriminator and the generator, respectively. In (5), we only consider a single-step update for $w_k$. This is a special case of Algorithm 1 in [1] with the batch-size $m$ set to 1. We note that the analysis presented in this paper can be naturally extended to the mini-batch case where $m$ is a finite number.

**Example 1.** We define $F(\widehat{D}(x)) = \widetilde{F}(\widehat{D}(x)) = x^2/2$, and the regularizer function $H(A) = \log \cosh(A - I)$, where $I$ is the identity matrix with the same dimension of $A$, and the function $\log \cosh(\cdot)$ transforms the input matrix element-wisely. We use this specific regularizer to control the magnitude of the model parameters $V$ and $w$. In practice, any convex function with its minimum reached at zero would be fine. Our choice $\log \cosh(A - I)$ here is is just a convenient special case since its derivative $H'(x) = \tanh(x)$ is smooth and bounded. Furthermore, we set the regularization parameter $\lambda \to \infty$, the original problem (3) becomes a constrained MinMax problem

$$\min_{\mathrm{diag}(V^\top V) = I_d} \max_{\|w\| = 1} \mathbb{E}_{y \sim \mathcal{P}} \mathbb{E}_{\widetilde{y} \sim \widetilde{\mathcal{P}}} \left[ (y^\top w)^2 - (\widetilde{y}^\top w)^2 \right],$$

in which the diagonal operation $\mathrm{diag}(\boldsymbol{A})$ returns a matrix where the diagonal entries are the same as $\boldsymbol{A}$ and the off-diagonal entries are all zero. The condition $\mathrm{diag}(\boldsymbol{V}^\top \boldsymbol{V}) = \boldsymbol{I}_d$ ensures that each column vector of $\boldsymbol{V}$ is normalized.

## 3 Dynamics of the GAN

**Definition 1.** Let $\boldsymbol{X}_k \overset{\text{def}}{=} [\boldsymbol{U}, \boldsymbol{V}_k, \boldsymbol{w}_k] \in \mathbb{R}^{n \times (2d+1)}$. We call $\boldsymbol{X}_k$ the *microscopic state* of the training process at iteration step $k$.

The microscopic state $\boldsymbol{X}_k$ contains all the information about the training process. In fact, the sequence $\{\boldsymbol{X}_k\}_{k=0,1,2,\ldots}$ forms a Markov chain on $\mathbb{R}^{n \times (2d+1)}$. This can be easily verified from the update rule of $\boldsymbol{X}_k$ as defined in (5), in which the real data $\boldsymbol{y}_k$ and fake data $\widetilde{\boldsymbol{y}}_k$ are drawn according to (1) and (2) respectively. The Markov chain is driven by the initial state $\boldsymbol{X}_0$ and the sequence of random variables $\{(\boldsymbol{c}_k, \boldsymbol{a}_k, \widetilde{\boldsymbol{c}}_{2k}, \widetilde{\boldsymbol{a}}_{2k}, \widetilde{\boldsymbol{c}}_{2k+1}, \widetilde{\boldsymbol{a}}_{2k+1})\}_{k=0,1,2,\ldots}$.

**Definition 2.** Let $\boldsymbol{P}_k \overset{\text{def}}{=} \boldsymbol{U}^\top \boldsymbol{V}_k$, $\boldsymbol{q}_k \overset{\text{def}}{=} \boldsymbol{U}^\top \boldsymbol{w}_k$, $\boldsymbol{r}_k \overset{\text{def}}{=} \boldsymbol{V}_k^\top \boldsymbol{w}_k$, $\boldsymbol{S}_k \overset{\text{def}}{=} \boldsymbol{V}_k^\top \boldsymbol{V}_k$, and $z_k \overset{\text{def}}{=} \boldsymbol{w}_k^\top \boldsymbol{w}_k$. We call the tuple $\{\boldsymbol{P}_k, \boldsymbol{q}_k, \boldsymbol{r}_k, \boldsymbol{S}_k, z_k\}$ the *macroscopic state* of the Markov chain $\boldsymbol{X}_k$ at step $k$.

Those macroscopic quantities measure the cosine similarities among the feature vectors of the true model $\boldsymbol{U}$, the generator $\boldsymbol{V}_k$ and the discriminator $\boldsymbol{w}_k$. For example, the cosine of the angle between the $i$th true feature (*i.e.*, the $i$th column of $\boldsymbol{U}$) and the $j$th feature estimated in the generator (*i.e.*, the $j$th column of $\boldsymbol{V}_k$) is $[\boldsymbol{P}_k]_{i,j}/\sqrt{[\boldsymbol{S}_k]_{j,j}}$, where $[\boldsymbol{P}_k]_{i,j}$ is the inner product between the two feature vectors and $\sqrt{[\boldsymbol{S}_k]_{j,j}}$ is the norm of the $j$th column of $\boldsymbol{V}_k$. (The columns of $\boldsymbol{U}$ are unit vectors and need not be normalized here.) For simplicity, we introduce a compact notation for the macroscopic state:

$$\boldsymbol{M}_k \overset{\text{def}}{=} \boldsymbol{X}_k^\top \boldsymbol{X}_k = \begin{bmatrix} \boldsymbol{I} & \boldsymbol{P}_k & \boldsymbol{q}_k \\ \boldsymbol{P}_k^\top & \boldsymbol{S}_k & \boldsymbol{r}_k \\ \boldsymbol{q}_k^\top & \boldsymbol{r}_k^\top & z_k \end{bmatrix}. \tag{6}$$

In what follows, we investigate the dynamics of the training algorithm (5) at both the macroscopic and the microscopic levels. At the macroscopic level, by examining the cosines of the angles, we study how closely the model parameters $\boldsymbol{V}_k$, $\boldsymbol{w}_k$ associated with the generator and discriminator can align with the ground truth feature vectors, *i.e.,* the columns of $\boldsymbol{U}$. At the microscopic level, we study how the elements in the matrix $\boldsymbol{V}_k$ and the vector $\boldsymbol{w}_k$ evolve as a stochastic process. As our analysis will reveal, the mechanisms behind the two levels are different: the macroscopic dynamics is asymptotically deterministic whereas the microscopic dynamics stays stochastic even as $n \to \infty$.

### 3.1 Macroscopic dynamics

We first study the asymptotic dynamics of the macroscopic state $\boldsymbol{M}_k$. Our theoretical analysis is carried out under the following assumptions.

(A.1) The sequences of $\boldsymbol{c}_k \sim \mathcal{P}_{\boldsymbol{c}}$ and $\widetilde{\boldsymbol{c}}_k \sim \mathcal{P}_{\widetilde{\boldsymbol{c}}}$ for $k = 0, 1, \ldots$ are i.i.d. random variables with bounded moments of all orders, and $\{\boldsymbol{c}_k\}$ is independent of $\{\widetilde{\boldsymbol{c}}_k\}$.

(A.2) The sequences $\{\boldsymbol{a}_k\}$ and $\{\widetilde{\boldsymbol{a}}_k\}$ for $k = 0, 1, \ldots$ are both independent Gaussian vectors with zero mean and the covariance matrix $\boldsymbol{I}_n$. Moreover, $\{\boldsymbol{a}_k\}$, $\{\widetilde{\boldsymbol{a}}_k\}$ are independent of $\{\boldsymbol{c}_k\}$ and $\{\widetilde{\boldsymbol{c}}_k\}$.

(A.3) The first-order derivative of $H(\cdot)$ and the derivatives up to fourth order of the functions $F(\widehat{D}(\cdot))$ and $\widetilde{F}(\widehat{D}(\cdot))$ exist and they are also uniformly bounded.

(A.4) Let $[\boldsymbol{U}, \boldsymbol{V}_0, \boldsymbol{w}_0]$ be the initial microscopic state. For $i = 1, 2, \ldots, n$, we have $\mathbb{E}\left[\sum_{\ell=1}^d ([\boldsymbol{U}]_{i,\ell}^4 + [\boldsymbol{V}_0]_{i,\ell}^4 + [\boldsymbol{w}_0]_i^4)\right] \leq C/n^2$, where $C$ is a constant not depending on $n$.

(A.5) The initial macroscopic state $\boldsymbol{M}_0$ satisfies $\mathbb{E}\|\boldsymbol{M}_0 - \boldsymbol{M}_0^*\| \leq C/\sqrt{n}$, where $\boldsymbol{M}_0^*$ is a deterministic matrix and $C$ is a constant not depending on $n$.

We provide a few remarks on the above assumptions. In Assumption (A.1), $\mathcal{P}_{\boldsymbol{c}}$ and $\mathcal{P}_{\widetilde{\boldsymbol{c}}}$ can be different. For example, $\boldsymbol{c}$ is Gaussian, and $\widetilde{\boldsymbol{c}}$ is uniform on $[-1, 1]^d$. The assumption (A.2) can

be relaxed to non-Gaussian cases as long as all moments of $\boldsymbol{a}_k$ and $\widetilde{\boldsymbol{a}}_k$ are bounded, but we use Gaussian assumption here to simplify the proof. The assumption (A.4) requires that the elements in the parameter matrix of real data $\boldsymbol{U}$ and initial microscopic state $\boldsymbol{X}_0$ are $\mathcal{O}(1/\sqrt{n})$ numbers. Intuitively, this assumption ensures that $\boldsymbol{U}$ and $\boldsymbol{X}_0$ are generic matrices with $\mathcal{O}(1)$ Frobenius norms (*i.e.,* not the matrices that most elements are zeros and only few elements are large numbers). The assumption (A.5) ensures that the initial macroscopic states converges to a deterministic value as the system size $n$ goes to infinity. The following theorem proves that if the initial state is convergent, then the whole training process converges to a deterministic process as $n \to \infty$, which is characterized by an ODE.

**Theorem 1.** *Fix $T > 0$. It holds under Assumptions (A.1)–(A.5) that*

$$\max_{0 \le k \le nT} \mathbb{E} \left\| \boldsymbol{M}_k - \boldsymbol{M}\left(\tfrac{k}{n}\right) \right\| \le \frac{C(T)}{\sqrt{n}}, \tag{7}$$

*where $C(T)$ is a constant that depends on $T$ but not on $n$, and* $\boldsymbol{M}(t) = \begin{bmatrix} \boldsymbol{I} & \boldsymbol{P}_t & \boldsymbol{q}_t \\ \boldsymbol{P}_t^\top & \boldsymbol{S}_t & \boldsymbol{r}_t \\ \boldsymbol{q}_t^\top & \boldsymbol{r}_t^\top & z_t \end{bmatrix} \in$

$\mathbb{R}^{(2d+1)\times(2d+1)}$ *is a deterministic function. Moreover, $\boldsymbol{M}(t)$ is the unique solution of the following ODE:*

$$
\begin{aligned}
\tfrac{\mathrm{d}}{\mathrm{d}t}\boldsymbol{P}_t &= \widetilde{\tau}\big(\boldsymbol{q}_t \widetilde{\boldsymbol{g}}_t^\top + \boldsymbol{P}_t \boldsymbol{L}_t\big) \\
\tfrac{\mathrm{d}}{\mathrm{d}t}\boldsymbol{q}_t &= \tau\big(\boldsymbol{g}_t - \boldsymbol{P}_t \widetilde{\boldsymbol{g}}_t + \boldsymbol{q}_t h_t\big) \\
\tfrac{\mathrm{d}}{\mathrm{d}t}\boldsymbol{r}_t &= \tau\big(\boldsymbol{P}_t^T \boldsymbol{g}_t - \boldsymbol{S}_t \widetilde{\boldsymbol{g}}_t + \boldsymbol{r}_t h_t\big) + \widetilde{\tau}\big(z_t \widetilde{\boldsymbol{g}}_t + \boldsymbol{L}_t \boldsymbol{r}_t\big) \\
\tfrac{\mathrm{d}}{\mathrm{d}t}\boldsymbol{S}_t &= \widetilde{\tau}\big(\boldsymbol{r}_t \widetilde{\boldsymbol{g}}_t^\top + \widetilde{\boldsymbol{g}}_t \boldsymbol{r}_t^\top + \boldsymbol{S}_t \boldsymbol{L}_t + \boldsymbol{L}_t \boldsymbol{S}_t\big) \\
\tfrac{\mathrm{d}}{\mathrm{d}t}z_t &= 2\tau\big(\boldsymbol{q}_t^\top \boldsymbol{g}_t - \boldsymbol{r}_t^\top \widetilde{\boldsymbol{g}}_t + z_t h_t\big) + \tau^2 b_t
\end{aligned}
\tag{8}
$$

*with the initial condition $\boldsymbol{M}(0) = \boldsymbol{M}_0^*$, where*

$$
\begin{aligned}
\boldsymbol{g}_t &= \big\langle \boldsymbol{c} f(\boldsymbol{c}^\top \boldsymbol{q}_t + e\sqrt{z_t \eta_T}) \big\rangle_{\boldsymbol{c},e}, \quad \widetilde{\boldsymbol{g}}_t = \big\langle \widetilde{\boldsymbol{c}} \widetilde{f}(\widetilde{\boldsymbol{c}}^\top \boldsymbol{r}_t + e\sqrt{z_t \eta_G}) \big\rangle_{\widetilde{\boldsymbol{c}},e}, \quad \boldsymbol{L}_t = -\lambda diag(H'(\boldsymbol{S}_t)) \\
h_t &= \big\langle f'(\boldsymbol{c}^\top \boldsymbol{q}_t + e\sqrt{z_t \eta_T}) \big\rangle_{\boldsymbol{c},e} - \big\langle \widetilde{f}'(\widetilde{\boldsymbol{c}}^\top \boldsymbol{r}_t + e\sqrt{z_t \eta_G}) \big\rangle_{\widetilde{\boldsymbol{c}},e} - \lambda H'(z_t), \\
b_t &= \eta_T \big\langle f^2(\boldsymbol{c}^\top \boldsymbol{q}_t + e\sqrt{z_t \eta_T}) \big\rangle_{\boldsymbol{c},e} + \eta_G \big\langle \widetilde{f}^2(\widetilde{\boldsymbol{c}}^\top \boldsymbol{r}_t + e\sqrt{z_t \eta_G}) \big\rangle_{\widetilde{\boldsymbol{c}},e}.
\end{aligned}
\tag{9}
$$

*The two functions $f, \widetilde{f}$ stand for $f(x) = \frac{\mathrm{d}}{\mathrm{d}x} F(\widehat{D}(x))$ and $\widetilde{f}(x) = \frac{\mathrm{d}}{\mathrm{d}x} \widetilde{F}(\widehat{D}(x))$, and $f', \widetilde{f}'$ and $H'$ are derivatives of $f, \widetilde{f}$ and $H$ respectively. The two constants $\eta_T$ and $\eta_G$ are the strength of the noise in the true data model and the generator, respectively. The brackets $\langle \cdot \rangle_{\boldsymbol{c},e}$ and $\langle \cdot \rangle_{\widetilde{\boldsymbol{c}},e}$ denote the averages over the random variables $\boldsymbol{c} \sim \mathcal{P}_{\boldsymbol{c}}$, $\widetilde{\boldsymbol{c}} \sim \mathcal{P}_{\widetilde{\boldsymbol{c}}}$, and $e \sim \mathcal{N}(0,1)$, where $\mathcal{P}_{\boldsymbol{c}}$ and $\mathcal{P}_{\widetilde{\boldsymbol{c}}}$ are the distributions involved in defining the generative model (1) and the generator (2).*

This theorem implies that for each $k = \lfloor tn \rfloor$ for some $t \in [0, T]$, the macroscopic state $\boldsymbol{M}_k$ converges to a deterministic number $\boldsymbol{M}(t)$, and the convergence rate is $\mathcal{O}(1/\sqrt{n})$. The limiting ODE (8) for the macroscopic states involves $\mathcal{O}(d^2)$ variables, where $d$ is the number of internal features often assumed to be a finite number that is much less than $n$. This ODE is essentially different from the ODE derived in the small-learning-rate limit [11–16], in which the number of variables is $\mathcal{O}(n)$.

The complete proof can be found in the Supplementary Materials. We briefly sketch the proof here. First, we note that $\boldsymbol{M}_k$ is a discrete-time stochastic process driven by the Markov chain $\boldsymbol{X}_k$. Then, we apply the martingale decomposition for $\boldsymbol{M}_k$ and get

$$\boldsymbol{M}_{k+1} - \boldsymbol{M}_k = \tfrac{1}{n}\phi(\boldsymbol{M}_k) + (\boldsymbol{M}_{k+1} - \mathbb{E}_k \boldsymbol{M}_{k+1}) + [\mathbb{E}_k \boldsymbol{M}_{k+1} - \boldsymbol{M}_k - \tfrac{1}{n}\phi(\boldsymbol{M}_k)],$$

where the matrix-valued function $\phi(\boldsymbol{M})$ represents the functions on the right hand sides of the ODE (8), and $\mathbb{E}_k$ denotes the conditional expectation given the state of the Markov chain $\boldsymbol{X}_k$. Finally, we show the martingale $\sum_{k'=0}^{k}(\boldsymbol{M}_{k'+1} - \mathbb{E}_{k'}\boldsymbol{M}_{k'})$ and the higher-order term $\mathbb{E}_k \boldsymbol{M}_{k+1} - \boldsymbol{M}_k - \tfrac{1}{n}\phi(\boldsymbol{M}_k)$ have no contribution when $n$ goes to infinity.

Due to the limitation of our current proof, the constant $C(T)$ in (7) grows exponentially as $T$ increases. This is not a problem for any finite $T$, but may cause some problem to study the long time behavior when $T \to \infty$. However, if we impose a sufficient large regularizer parameter $\lambda$ to limit the norms of the microscopic weights $\boldsymbol{V}_k$ and $\boldsymbol{w}_k$, then the macroscopic state $\boldsymbol{M}_k$ is bounded

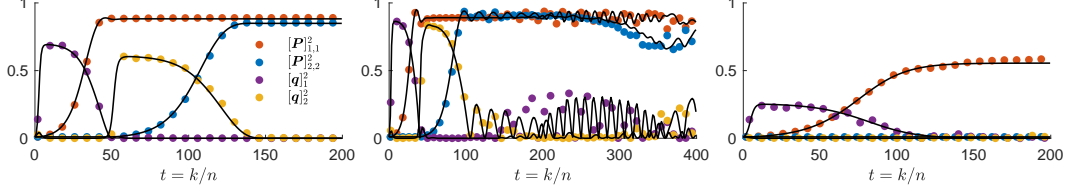

Figure 1: Macroscopic dynamics of the GAN with $d = 2$ features: $[\boldsymbol{P}_k]_{i,j}$ is the cosine of the angle between $i$'th column vector of the real feature matrix $\boldsymbol{U}_k$ and $j$'th column vector of the generator's weight matrix $\boldsymbol{V}_k$. Similarly, $[\boldsymbol{q}_k]_i$ is the cosine of angle between $i$'th column vector of $\boldsymbol{U}_k$ and the discriminator's weight vector $\boldsymbol{w}_k$. Colored dots are results from experiments, and the curves tracing these dots are our theoretical prediction by the ODE (8). From the left to right, the variance of background noise is $\eta_{\mathrm{T}} = \eta_{\mathrm{G}} = 2, 1, 4$ respectively, and other parameters are the same. The left figure is an example of successful training, where two features (red and blue dots) are retrieved by the generator. The center figure shows an oscillating training. It happens when noise are weak. The right figures shows a mode collapsing state, in which only the first feature are estimated by the generator.

as $[\boldsymbol{M}_k]^2_{i,j} \leq [\boldsymbol{M}_k]_{i,i}[\boldsymbol{M}_k]_{j,j}$. In our experiments, $\lambda > 1$ is sufficient. In this case, the constant $C(T)$ is bounded not depending on $T$. In Example 1, when $\lambda \to \infty$, $[\boldsymbol{M}_k]_{i,i} = 1$, and therefore $[\boldsymbol{M}_k]^2_{i,j} \leq 1$ and $C(T) \leq (2d + 1)^2$, where the number of features $d$ is considered a constant not growing with $n$. This justifies the fixed points analysis of the ODE as discussed in Section 4, which reflects the long-time training behavior. A better proof strategy to get rid of this dependence of $T$ is also possible, *e.g.*, [26].

**Numerical verification.** We verify the theoretical prediction given by the ODE (8) via numerical simulations under the settings stated in Example 1. The results are shown in Figure 1. The number of features is $d = 2$, and $\boldsymbol{c}_k$ and $\widetilde{\boldsymbol{c}}_k$ are both Gaussian with zero mean and covariance diag$([5, 3])$. The dimension is $n = 5,000$, and the learning rates of the generator and discriminator are $\widetilde{\tau} = 0.04$ and $\tau = 0.2$ respectively. After testing different noise strength $\eta_{\mathrm{T}} = \eta_{\mathrm{G}} = 2, 1, 4$, we have observed at least three nontrivial dynamical patterns: success, oscillating or mode collapsing. In all these experiments, our theoretical predictions match the actual trajectories of the macroscopic states pretty well.

Let us take a closer look at the successful case as shown in the left figure in Figure 1. The dynamics can be split into 4 stages. At the first stage, the discriminator learns the first feature of the true model. At this state, $[\boldsymbol{q}_t]_1$ quickly increases. At the second stage, the generator starts to learn the first feature and the discriminator is deceived. At this stage, $[\boldsymbol{P}_t]^2_{1,1}$ increases and $[\boldsymbol{q}_t]^2_1$ decreases. Once the discriminator completely forgets the first feature as $[\boldsymbol{q}_t]_1 \approx 0$, the third state begins. The discriminator starts to learn the second feature as $[\boldsymbol{q}_t]^2_2$ increases. Then, at the last stage, the generator learns the second feature and the discriminator is fooled again. In this region, $[\boldsymbol{P}_t]^2_{2,2}$ increases and $[\boldsymbol{q}_t]^2_2$ decreases down to 0. Eventually, the generators learns both features and the discriminator is completely fooled. It ends up at a stationary state that $\boldsymbol{q}_t = \boldsymbol{0}$ and $\boldsymbol{P}_t$ is nearly an identity matrix. Interestingly, this experiment shows that the generator learn features sequentially given a single-feature discriminator. This may be a reason why in practice, the discriminator's structure can be much simpler than the generator's.

## 3.2 Microscopic dynamics

In this section, we study how the elements in $\boldsymbol{X}_k = [\boldsymbol{U}, \boldsymbol{V}_k, \boldsymbol{w}_k]$ evolve during the training process. Instead of studying the trajectory of $\boldsymbol{X}_k$, we study the evolution of the *empirical measure* of the microscopic states, which is defined as

$$\mu_k(\widehat{\boldsymbol{u}}, \widehat{\boldsymbol{v}}, \widehat{w}) \stackrel{\text{def}}{=} \tfrac{1}{n} \sum_{i=1}^n \delta\big([\widehat{\boldsymbol{u}}^\top, \widehat{\boldsymbol{v}}^\top, \widehat{w}] - \sqrt{n}\big[[\boldsymbol{U}]_{i,:}, [\boldsymbol{V}_k]_{i,:}, [\boldsymbol{w}]_i\big]\big)$$

where $\delta(\cdot)$ is a Dirac measure on $\mathbb{R}^{2d+1}$ and $[\boldsymbol{U}]_{i,:}, [\boldsymbol{V}_k]_{i,:}$ are $i$th row of $\boldsymbol{U}$ and $\boldsymbol{V}_k$ respectively. The scaling factor $\sqrt{n}$ in the Dirac measures is introduced because $[\boldsymbol{U}]_{i,\ell}, [\boldsymbol{V}_k]_{i,\ell}$ and $[w_{k,}]_i$ are $\mathcal{O}(1/\sqrt{n})$ quantities.

We next embed the discrete-time measure-valued stochastic process $\mu_k$ into a continuous-time process by defining $\mu_t^{(n)} \stackrel{\text{def}}{=} \mu_k(\widehat{\boldsymbol{u}}, \widehat{\boldsymbol{v}}, \widehat{w})$ with $k = \lfloor nt \rfloor$. Following the general technical approach presented in [17], we can show that under the same assumptions as Theorem 1, given $T > 0$, the sequence of measure-valued process $\{\{\mu_t^{(n)}\}_{t \in [0,T]}\}_n$ converges weakly to a deterministic process $\{\mu_t\}_{t \in [0,T]}$. In addition, $\mu_t$ is the measure of the solution to the stochastic differential equation

$$
\begin{aligned}
\mathrm{d}\widehat{\boldsymbol{u}}_t &= 0 \\
\mathrm{d}\widehat{\boldsymbol{v}}_t &= \widetilde{\tau}\big(\widehat{w}_t \widetilde{\boldsymbol{g}}_t + \boldsymbol{L}_t \widehat{\boldsymbol{v}}_t\big)\,\mathrm{d}t \\
\mathrm{d}\widehat{w}_t &= \tau\big(\widehat{\boldsymbol{u}}_t^\top \boldsymbol{g}_t + \widehat{\boldsymbol{v}}_t^\top \widetilde{\boldsymbol{g}}_t + \widehat{w}_t h_t\big)\,\mathrm{d}t + \tau\sqrt{b_t}\,\mathrm{d}B_t
\end{aligned}
\tag{10}
$$

where $(\widehat{\boldsymbol{u}}_0, \widehat{\boldsymbol{v}}_0, \widehat{w}_0) \sim \mu_0$; $B_t$ is the standard Brownian motion. The functions $\boldsymbol{g}_t$, $\widetilde{\boldsymbol{g}}_t$, $\boldsymbol{L}_t$, $h_t$ and $b_t$ are defined in (9), in which the macroscopic quantities $\boldsymbol{P}_t$, $\boldsymbol{S}_t$, $\boldsymbol{q}_t$, $z_t$, $\boldsymbol{r}_t$ are computed as follows

$$
\boldsymbol{P}_t = \langle \mu_t, \widehat{\boldsymbol{u}}\widehat{\boldsymbol{v}}^\top \rangle, \quad \boldsymbol{S}_t = \langle \mu_t, \widehat{\boldsymbol{v}}\widehat{\boldsymbol{v}}^\top \rangle, \boldsymbol{q}_t = \langle \mu_t, \widehat{\boldsymbol{u}}\widehat{w} \rangle, \quad z_t = \langle \mu_t, \widehat{w}^2 \rangle, \quad \boldsymbol{r}_t = \langle \mu_t, \widehat{\boldsymbol{v}}\widehat{w} \rangle, \tag{11}
$$

where $\langle \mu_t, \cdot \rangle$ denotes the expectation with respect to the measure $\mu_t$.

The SDE (10) shows the intuitive meaning of the functions defined in (9): $\boldsymbol{g}_t$, $\widetilde{\boldsymbol{g}}_t$, $\boldsymbol{L}_t$, $h_t$ are drift coefficients of the SDE and $b_t$ is the diffusion coefficient of the SDE. We also note that if one follows the analysis in the small-learning-rate limit [11–16], one will get an ODE for the microscopic states. Compared to our SDE formula, the diffusion term $\tau\sqrt{b_t}dB_t$ is missing in those works, and therefore the effect of the noise can not be analyzed.

Moreover, the deterministic measure $\mu_t$ is unique solution of the following PDE (given in its weak form): for any bounded smooth test function $\varphi(\widehat{\boldsymbol{u}}, \widehat{\boldsymbol{v}}, \widehat{w})$,

$$
\begin{aligned}
\frac{\mathrm{d}}{\mathrm{d}t}\big\langle \mu_t, \varphi(\widehat{\boldsymbol{u}}, \widehat{\boldsymbol{v}}, \widehat{w}) \big\rangle = \\
\widetilde{\tau}\big\langle \mu_t, \big(\widehat{w}\widetilde{\boldsymbol{g}}_t^\top + \widehat{\boldsymbol{v}}^\top \boldsymbol{L}_t\big)\nabla_{\widehat{\boldsymbol{v}}}\varphi \big\rangle + \tau\big\langle \mu_t, \big(\widehat{\boldsymbol{u}}^\top \boldsymbol{g}_t - \widehat{\boldsymbol{v}}^\top \widetilde{\boldsymbol{g}}_t + h_t\widehat{w}\big)\frac{\partial}{\partial \widehat{w}}\varphi \big\rangle + \frac{\tau^2}{2}b_t\big\langle \mu_t, \frac{\partial^2}{\partial \widehat{w}^2}\varphi \big\rangle
\end{aligned}
\tag{12}
$$

where $\boldsymbol{q}_t$, $\boldsymbol{r}_t$, $\boldsymbol{S}_t$, and $z_t$ are defined in (11), and the functions $\boldsymbol{g}_t$, $\widetilde{\boldsymbol{g}}_t$, $b_t$, $h_t$ and $\boldsymbol{L}_t$ are defined in (9). We refer readers to [17] for a general framework for rigorously establishing the above scaling limit.

The connection between the microscopic and macroscopic dynamics can also be derived from the weak formulation of the PDE. Let $\varphi$ being each element of $\widehat{\boldsymbol{u}}\widehat{\boldsymbol{v}}^\top$, $\widehat{\boldsymbol{u}}\widehat{w}$, $\widehat{\boldsymbol{v}}\widehat{w}$, $\widehat{\boldsymbol{v}}\widehat{\boldsymbol{v}}^\top$, $\widehat{w}^2$, and substituting those $\varphi$ into the PDE (12), we can derive the ODE (8). In the setting of this paper, the macroscopic dynamics enjoys a closed ODE: We can predict the macroscopic states without solving the PDE nor SDE at microscopic scale. However, in a more general setting, e.g. when we add a regularizer other than the L2 type, the ODE itself may not be closed. In that case, one has to solve the PDE directly.

**Numerical verification.** We verify the predictions given by the PDE (12) by setting $d = 1$ using a special choice of the $(n \times 1)$-dimensional target feature matrix $\boldsymbol{U}$ whose elements are all $1/\sqrt{n}$ with $n = 10,000$. We also set the initial condition $\mu_0(\widehat{v}, \widehat{w}|\widehat{u} = 1)$ to be a Gaussian distribution. (When $d = 1$, the macroscopic quantities $P_t, q_t, r_t, S_t$ reduce to scalars, so we remove their boldface here.) In this case, the PDE (12) admits a particularly simple analytical solution: at any time $t$, the solution $\mu_t(\widehat{v}, \widehat{w}|\widehat{u} = 1)$ is a Gaussian distribution whose mean and covariance matrix are given by

$\mathbb{E}_{\mu_t(\widehat{v},\widehat{w}|\widehat{u}=1)}\begin{bmatrix} \widehat{v} \\ \widehat{w} \end{bmatrix} = \begin{bmatrix} P_t \\ q_t \end{bmatrix}$, $\mathbb{E}_{\mu_t(\widehat{v},\widehat{w}|\widehat{u}=1)}\begin{bmatrix} \widehat{v} \\ \widehat{w} \end{bmatrix}\begin{bmatrix} \widehat{v} & \widehat{w} \end{bmatrix} = \begin{bmatrix} S_t & r_t \\ r_t & z_t \end{bmatrix}$. Figure 2 overlays the contours of the probability distribution $\mu_t(\widehat{v}, \widehat{w}|\widehat{u} = 1)$ at different times $t$ over the point clouds of the actual experiment data $(\sqrt{n}[\boldsymbol{w}_k]_i, \sqrt{n}[\boldsymbol{V}_k]_{i,1})$. We can see that the theoretical prediction given by (12) has excellent agreement with simulation results.

## 4 Local Stability Analysis of the ODE for the Macroscopic States

In this section, we study how the parameters, such as the learning rates $\tau$ and $\widetilde{\tau}$, noise strength $\eta_{\mathrm{G}}$ and $\eta_{\mathrm{T}}$ affect the training algorithm. We will focus on the concrete model as described in Example 1 so that we can have analytical solutions.

In order to further reduce the degrees of freedom of the ODE (8), we let the regularization parameter $\lambda \to \infty$. In this case, the vector $\boldsymbol{w}_k$ and all columns vectors of $\boldsymbol{V}_k$ are always normalized. Thus

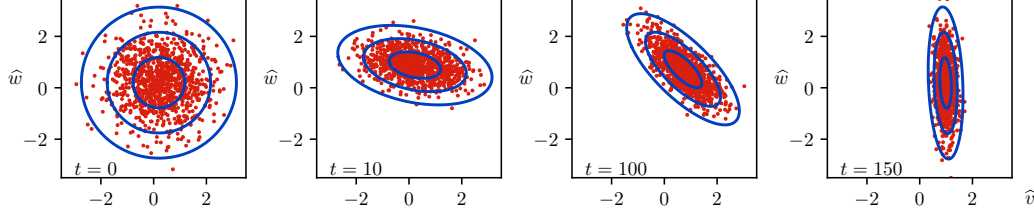

Figure 2: The evolution of the microscopic states at $t = 0, 10, 100$, and $150$. For each fixed $t$, the red points in the corresponding figure represent the values of $(\widehat{v}, \widehat{w}) = (\sqrt{n}[\boldsymbol{V}_k]_{i,1}, \sqrt{n}[\boldsymbol{w}_k]_i)$ for $i = 1, 2, \ldots, n$, where $k = \lfloor nt \rfloor$. The blue ellipses illustrate the contours corresponding to one, two, and three standard deviations of the 2-D Gaussian distribution predicted by the PDE (12).

$z_k = 1$ and $[\boldsymbol{S}]_{i,i} = 1$. The macroscopic state is then described by $\boldsymbol{P}_k, \boldsymbol{q}_k, \boldsymbol{r}_k$ and off-diagonal terms of $\boldsymbol{S}_k$. Correspondingly, the ODE in Theorem 1 reduces to

$$
\begin{cases}
\frac{\mathrm{d}}{\mathrm{d}t} \boldsymbol{P}_t & = \widetilde{\tau}\left( \boldsymbol{q}_t \boldsymbol{r}_t^{\top} \widetilde{\boldsymbol{\Lambda}} + \boldsymbol{P}_t \boldsymbol{L}_t \right) \\
\frac{\mathrm{d}}{\mathrm{d}t} \boldsymbol{q}_t & = \tau\left( \boldsymbol{\Lambda} \boldsymbol{q}_t - \boldsymbol{P}_t \widetilde{\boldsymbol{\Lambda}} \boldsymbol{r}_t + h_t \boldsymbol{q}_t \right) \\
\frac{\mathrm{d}}{\mathrm{d}t} \boldsymbol{r}_t & = \tau\left( \boldsymbol{P}_t^{T} \boldsymbol{\Lambda} \boldsymbol{q}_t - \boldsymbol{S}_t \widetilde{\boldsymbol{\Lambda}} \boldsymbol{r}_t + h_t \boldsymbol{r}_t \right) + \widetilde{\tau}\left( \widetilde{\boldsymbol{\Lambda}} + \boldsymbol{L}_t \right) \boldsymbol{r}_t \\
\frac{\mathrm{d}}{\mathrm{d}t} \boldsymbol{S}_t & = \widetilde{\tau}\left( \boldsymbol{r}_t \boldsymbol{r}_t^{\top} \widetilde{\boldsymbol{\Lambda}} + \widetilde{\boldsymbol{\Lambda}} \boldsymbol{r}_t \boldsymbol{r}_t^{\top} + \boldsymbol{S}_t \boldsymbol{L}_t + \boldsymbol{L}_t \boldsymbol{S}_t \right)
\end{cases}
\tag{13}
$$

where $\boldsymbol{\Lambda}$ and $\widetilde{\boldsymbol{\Lambda}}$ are the covariance matrices of the distributions $P_c$ and $P_{\widetilde{c}}$, respectively; and

$$
h_t = (1 - \tfrac{\tau \eta_G}{2}) \boldsymbol{r}_t^{\top} \widetilde{\boldsymbol{\Lambda}} \boldsymbol{r}_t - (1 + \tfrac{\tau \eta_T}{2}) \boldsymbol{q}_t^{\top} \boldsymbol{\Lambda} \boldsymbol{q}_t - \tau \tfrac{\eta_G^2 + \eta_T^2}{2}, \qquad \boldsymbol{L}_t = -\mathrm{diag}(\boldsymbol{r}_t \boldsymbol{r}_t^{\top} \widetilde{\boldsymbol{\Lambda}}),
\tag{14}
$$

in which $\eta_T$ and $\eta_G$ are the variance of noise in the true data model and generator, respectively. The derivation from the ODE (8) to (13) is presented in the Supplementary Materials.

Next, we discuss under what conditions, the GAN can reach a desirable training state by studying local stability of a particular type of fixed points of the ODE (13). The perfect estimation of the generator corresponds to $\boldsymbol{P}_t$ being an identity matrix (up to a permutation of rows and columns). A complete fail state relates to $\boldsymbol{P} = \boldsymbol{0}$. Furthermore, It is easy to verify that if $\boldsymbol{q}_t = \boldsymbol{r}_t = \boldsymbol{0}$, the ODE (13) will be stable for any $\boldsymbol{P}_t = \boldsymbol{P}$.

**Claim 1.** *The macroscopic states $\boldsymbol{P}_t$, $\boldsymbol{q} = \boldsymbol{r} = \boldsymbol{0}$ for all valid $\boldsymbol{P}_t$ are always the fixed points of the ODE (13). Furthermore, a sufficient condition that the perfect estimation state $\boldsymbol{P}_t = \boldsymbol{I}$, $\boldsymbol{q} = \boldsymbol{r} = \boldsymbol{0}$ is locally stable and the failed state $\boldsymbol{P}_t = \boldsymbol{0}$, $\boldsymbol{q} = \boldsymbol{r} = \boldsymbol{0}$ is unstable if*

$$
\tfrac{1}{2} \max_{\ell}\{ \Lambda_\ell - \widetilde{\Lambda}_\ell + \alpha \widetilde{\Lambda}_\ell \} \leq \tau \overline{\eta^2} < \min_{\ell} \Lambda_\ell,
\tag{15}
$$

*where $\alpha = \frac{\widetilde{\tau}}{\tau}$, $\overline{\eta^2} = \frac{1}{2}(\eta_T^2 + \eta_G^2)$, and $\Lambda_\ell = [\boldsymbol{\Lambda}]_{\ell,\ell}$, $\widetilde{\Lambda}_\ell = [\widetilde{\boldsymbol{\Lambda}}]_{\ell,\ell}$.*

The proof can be found in the Supplementary Materials. If the right inequality in (15) is violated, any feature $\ell$ with the signal-to-noise ratio $[\boldsymbol{\Lambda}]_{\ell,\ell} < \tau \overline{\eta^2}$ is not learned by the generator resulting *mode collapsing*. The right figure in Figure 1 demonstrates this situations, where only one of the two features is recovered. If the left inequality in (15) is violated, the training processes can be trapped in an *oscillation phase*. This phenomenon is shown in the middle figure in Figure 1. This result indicates that proper background noise can help to avoid oscillation and stabilize the training process. In fact, the trick of injecting additional noise has been used in practice to train multi-layer GANs [27]. To our best knowledge, our paper is the first theoretical study on why noise can have such a positive effect via a dynamic perspective.

In experiments, the training is not ended at the perfect recovery point due to the presence of the noise but converges at another fixed point nearby. This is because the perfect state is marginally stable, as the Jacobian matrix always has zero eigenvalues. It indicates that there are other locally stable fixed points near $\boldsymbol{P} = \boldsymbol{I}$. In fact, all points in the hyper-rectangle region satisfying $\boldsymbol{q} = \boldsymbol{r} = \boldsymbol{0}$ and $|p_\ell^*| \leq |[\boldsymbol{P}]_{\ell,\ell}| \leq 1$, $\forall \ell = 1, 2, \ldots, d$ are locally stable for some critical $p_\ell^*$. In the matched case when $\Lambda_\ell = \widetilde{\Lambda}_\ell$, we have $p_\ell^* = \left[ (\Lambda_\ell - \tau \overline{\eta^2})(\widetilde{\Lambda}_\ell + \tau \overline{\eta^2} - \alpha \widetilde{\Lambda}_\ell)/(\Lambda_\ell \widetilde{\Lambda}_\ell) \right]^{1/2}$, $\alpha = \frac{\widetilde{\tau}}{\tau}$ and $\overline{\eta^2} =$

$\frac{1}{2}(\eta_{\mathrm{T}}^2 + \eta_{\mathrm{G}}^2)$. Starting from a point near the origin, numerical solution of the ODE shows the training processes are ended up at the corner of this hyper-rectangle, *i.e.*, $\boldsymbol{P}^* = \mathrm{diag}(\{p_\ell^*, \ \ell = 1, 2, \ldots, d\})$. In the small-learning rate limit $\tau \to 0$ and the learning rate ratio $\alpha \to 0$, we get the perfect recovery $\boldsymbol{P}^* = \boldsymbol{I}$. The limit $\tau \to 0$, $\alpha \to 0$ was studied in the small-learning-rate analysis with the two-time scaling [15], and the result is consistent, but our analysis includes the situations with finite $\tau$ and $\alpha$.

In addition, we provide a phase diagram analysis in a single-feature case $d = 1$ in the Supplementary Materials. All possible fixed points in this case are enumerated and their local stability is analyzed. This helps us understand the successful recovery condition (15), which is the intersection of the informative phases that each feature can be recovered individually.

## 5  Conclusion

We present a simple high-dimensional model for GAN with an exactly analyzable training process. Using the tool of scaling limits of stochastic processes, we show that the macroscopic state associated with the training process converges to a deterministic process characterized as the unique solution of an ODE, whereas the microscopic state remains stochastic described by an SDE, whose time-varying probability measure is described by a limiting PDE.

Indeed, it is a common picture in statistical physics that the macroscopic states of large systems tend to converge to deterministic values due to self-averaging. These notions, especially the mean-field dynamics, have been applied to analyzing neural networks both in shallow [19, 20] and deep models [28]. However, this mean-field regime was not considered in previous analyses of GAN. For example, a series of recent works *e.g.*, [11–16] considers a different scaling regime where the learning rate goes to zero but the system dimension $n$ stays fixed. In that regime, the microscopic dynamics are deterministic even with the presence of the microscopic noise. In contrast, we study the regime where the learning rate is fixed but the dimension $n \to \infty$. This setting allows us to quantify the effect of training noise in the learning dynamics.

In this paper, we only consider a linear generator with a latent variable $\widetilde{\boldsymbol{c}}$ drawn from a fixed distribution $\mathcal{P}_{\widetilde{\boldsymbol{c}}}$, but our analysis can be extended to a more complex non-linear model with a learnable latent-variable distribution. Specifically, in order to compute derivatives w.r.t. $\mathcal{P}_{\widetilde{\boldsymbol{c}}}$, the latent variable $\widetilde{\boldsymbol{c}} \sim \mathcal{P}_{\widetilde{\boldsymbol{c}}}$ should be reparameterized by a deterministic function $\widetilde{\boldsymbol{c}} = f(\boldsymbol{z}; \boldsymbol{\theta})$, where $\boldsymbol{\theta}$ is a learnable parameter and $\boldsymbol{z}$ is a random variable drawn from a simple and fixed distribution. For example, a Gaussian mixture with $L$ equal-probability modes can be parameterized by $\widetilde{\boldsymbol{c}} = \sum_{\ell=1}^{L}(\boldsymbol{\mu}_\ell + \boldsymbol{\Sigma}_\ell \boldsymbol{\epsilon}_l)\beta_l$, where $\boldsymbol{\mu}_\ell$ and $\boldsymbol{\Sigma}_\ell$ are two learnable parameters representing the mean and covariance of the $\ell$th mode respectively, and $\boldsymbol{\epsilon} \sim \mathcal{N}(0, \boldsymbol{I})$; $\beta_\ell$ is a random indicator variable where only one $\beta_\ell$ for $\ell = 1, 2, \ldots, L$ is 1 and the others are 0. In practice, $f(\boldsymbol{z}; \theta)$ is implemented by a multilayer neural network. Our analysis can be naturally extended to analyzing this model as long as the dimensions of $\widetilde{\boldsymbol{c}}$ and $\boldsymbol{\theta}$ keep finite when the data dimension $n$ goes to infinity. More challenging situations, where the dimension of $\boldsymbol{\theta}$ is proportional to $n$, will be explored in future works.

Although our analysis is carried out in the asymptotic setting, numerical experiments show that our theoretical predictions can accurately capture the actual performance of the training algorithm at moderate dimensions. Our analysis also reveals several different phases of the training process that highly depend on the choice of the learning rates and noise strength. The analysis reveals a condition on the learning rates and the strength of noise to have successful training. Violating this condition results either oscillation or mode collapsing. Despite its simplicity, the proposed model of GAN provides a new perspective and some insights for the study of more realistic models and more involved training algorithms.

**Acknowledgments**  This work was supported by the US Army Research Office under contract W911NF-16-1-0265 and by the US National Science Foundation under grants CCF-1319140, CCF-1718698, and CCF-1910410.

## Footnotes

[1]If $\boldsymbol{U}$ is not orthogonal, we can rewrite $\boldsymbol{U}\boldsymbol{c}$ in (1) as $(\boldsymbol{U}\boldsymbol{R})(\boldsymbol{R}^{-1}\boldsymbol{c})$, where $\boldsymbol{R}$ is a matrix that orthogonalizes and normalizes the columns of $\boldsymbol{U}$. We can then study an equivalent system where the new feature vector is $\boldsymbol{R}^{-1}\boldsymbol{c}$.

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
