[Supplementary Material]

# Supplementary Materials: A Solvable High-Dimensional Model of GAN

**Chuang Wang**[1,2]
wangchuang@ia.ac.cn

**Hong Hu**[2]
honghu@g.harvard.edu

**Yue M. Lu**[2]
yuelu@seas.harvard.edu

1. National Laboratory of Pattern Recognition, Institute of Automation,
Chinese Academy of Science, 95 Zhong Guan Cun Dong Lu, Beijing 100190, China
2. John A. Paulson School of Engineering and Applied Sciences, Harvard University
33 Oxford Street, Cambridge, MA 02138, USA

These Supplementary Materials provide additional information, detailed derivations and proof of the results shown in the main text. Specifically, in Section S-I we provide a local stability analysis and draw the phase diagram in the case $d = 1$ and $d = 2$. In Section S-II, we present a heuristic derivation of the stochastic differential equation (SDE) for the microscopic states. Next, in Section S-III, we show a derivation of the ODE for the macroscopic states from the weak formulation of the PDE. We then establish the full proof of the Theorem 1 in Section S-IV. Finally, we present the local stability analysis of the ODE's fixed points in Section S-V.

*Notation*: Throughout the paper, we use $\boldsymbol{I}_d$ to denote the $d \times d$ identity matrix. Depending on the context, $\|\cdot\|$ denotes either the $\ell_2$ norm of a vector or the spectral norm of a matrix. For any $x \in \mathbb{R}$, the floor operation $\lfloor x \rfloor$ gives the largest integer that is smaller than or equal to $x$. We denote $[\boldsymbol{v}]_i$ the $i$th element of the vector $\boldsymbol{v}$ and denote $[\boldsymbol{M}]_{i,j}$ the element at $i$th row and $j$th column of the matrix $\boldsymbol{M}$. Finally, $C(T)$ denotes a constant that depends on the terminal time $T$, and $C$ denotes a general constant that does not depends on $T$ and $n$. Both $C$ and $C(T)$ can vary line to line.

## S-I  Phase diagram for the case $d = 1$ and $d = 2$

In what follows, we provide a thorough study of all the fixed points of the ODE (13) when the number of feature $d = 1$ and $d = 2$. In particular, three major phases are identified under different settings of the learning rates $\tau$ and $\widetilde{\tau}$ with the fixed model parameters $\eta_{\mathrm{T}}$, $\eta_{\mathrm{G}}$, $\boldsymbol{\Lambda}$, and $\widetilde{\boldsymbol{\Lambda}}$ .

**Phase diagram for $d = 1$.**  By analyzing the local stabilities of these fixed points as illustrated in Figure 1(a), we obtain the phase diagram as shown in Figure 1(b). For simplicity, we only present the result when $\eta_{\mathrm{T}} = \eta_{\mathrm{G}} = 1$, and $\boldsymbol{\Lambda} = \widetilde{\boldsymbol{\Lambda}}$, which is denoted by $\Lambda$ used in the remaining part of this section. Detailed derivations are presented in S-V.

Even in this simplest case, we find there are in total 5 types of fixed points, the locations of which are visualized in the 3-dimensional space $(P, q, r)$ shown in Figure 1(a). Each type of the fixed points has an intuitive meaning in terms of the two-player game between $\mathcal{G}$ and $\mathcal{D}$. We list the detailed information in Table 1, in which we define a function $\beta(\tau) = \begin{cases} [1 + (\frac{\Lambda}{2} - \frac{\Lambda}{\tau})^{-1}]^{-1}, & \text{if } \tau \leq \frac{2\Lambda}{\Lambda+2} \\ +\infty, & \text{otherwise} \end{cases}$.

*Noninformative phase:* We say that the ODE (13) is in a noninformative phase if either a type-1 or type-2 fixed point in Table 1 is stable. In this case, $P = 0$, which indicates that the generator's parameter vector $\boldsymbol{V}$ has no correlation with the true feature vector $\boldsymbol{U}$. In Figure 1(b), the region labeled as noninfo-1 is the stable region for the type-1 fixed point, and noninfo-2 is the stable region for the type-2 fixed point. The two regions have no overlap. However, we note that in noninfo-1, the

Table 1: List of the fixed points of the ODE (13) when $d = 1$ and $\Lambda = \widetilde{\Lambda}$.

| Type | Location | Existence | Stable Region | Intuitive Interpretation |
|------|----------|-----------|---------------|--------------------------|
| 1 | $P = q = 0,$ $r = 0$ | always | $\tau > \Lambda^2, \frac{\widetilde{\tau}}{\tau} < \frac{\tau + \Lambda}{\Lambda}$ | Both $\mathcal{G}$ and $\mathcal{D}$ fail, and they are uncorrelated |
| 2 | $P = q = 0$ $r = \pm r^* \neq 0$ | $\frac{\widetilde{\tau}}{\tau} \geq \frac{\tau + \Lambda}{\Lambda}$ or $\frac{\widetilde{\tau}}{\tau} \leq 1 - \frac{\tau}{2}$ | $\max\{2, \frac{\tau + \Lambda}{\Lambda}\} \leq \frac{\widetilde{\tau}}{\tau} \leq \beta(\tau)$ | Both $\mathcal{G}$ and $\mathcal{D}$ fail, and they are correlated |
| 3 | $q = r = 0$ $|P| \in (0, 1]$ | always | $|P| = 1$ is stable if $\frac{\widetilde{\tau}}{\tau} \leq \min\{\frac{2\tau}{\Lambda}, \max\{\frac{\tau^2\Lambda^{-1}}{|\tau - \Lambda|}, 4\}\}$ | $\mathcal{G}$ wins and $\mathcal{D}$ loses |
| 4 | $P = r = 0$ $q = \pm q^* \neq 0$ | always | always unstable | $\mathcal{G}$ loses and $\mathcal{D}$ wins |
| 5 | None of $P, q$ or $r$ is zero | not always, at most 8 fixed points | can be computed numerically | Both $\mathcal{G}$ and $\mathcal{D}$ are informative |

type-3 fixed points can also be stable, in which case the stationary point of the ODE is determined by the initial condition.

*Informative phase:* We say that the ODE (13) is in an informative phase if neither type-1 nor type-2 fixed point is stable, and if at least one fixed point of type-3 and type-5 is stable. In this case, it is guaranteed that $P$ is nonzero, indicating that the generator can achieve non-vanishing correlation with the real feature vector. In addition, the stable regions for the type-3 and type-5 fixed points are disjoint. They are shown in Figure 1(b) as info-1 and info-2, respectively. The difference between the two region is that, in info-1, $q$ is exactly $0$ indicating that the discriminator is completely fooled, whereas in info-2, $q$ is nonzero.

*Oscillating phase:* We say that the ODE (13) is in an oscillating phase if none of the fixed points in Table 1 is stable. In this phase, limiting cycles emerge and the system will oscillate on these cycles indefinitely. Moreover, we found two types of limiting cycles.

To further illustrate the phase transitions, we draw ODE trajectories and phase portraits in Figure 2 corresponding to different choices of the step sizes (from left to right, $\widetilde{\tau} = 0.03, 0.2, 0.4, 0.47$).

The two figures in the first column of Figure 2 show a case in the Info-1 phase. The bottom red dot in Figure 1.(b) represents this configuration of the step sizes, where $\widetilde{\tau}/\tau$ is small. The top figure of Figure 2.(a) shows the dynamics of $P_t$, $q_t$ and $r_t$, and the bottom figure shows the phase portrait on $P - q$ plane. Top figure of Figure 2.(a) shows an interesting phenomenon that dynamics are separated into two stages. At the first stage, $q_t$ (red dots, cosine similarity between the true feature vector and discriminator's estimation) increases drastically from 0 to some value near 1, while $P_t$ (blue dots, cosine similarity between the true feature vector and generator's estimation) almost doesn't change. Intuitively, at this stage, the discriminator learns the true model while the generator is unchanged. In the second stage, the generator start to fool the discriminator, where $|P_t|$ increases and $q_t$ decreases. In fact, these two-stage dynamics can be understood from the ODE (13): When $\tau/\tau$ is small, the process can be decomposed into two processes in different time scales. In particular, the discriminator is associated with the faster dynamics as $\tau \gg \widetilde{\tau}$, and the generator governs the slower dynamics. Figure 1 in the main text shows that this picture is still hold for multi-feature cases in the hierarchical dynamics.

The figures in the middle two columns of Figure 2 show the two types of limiting cycles that can emerge in the oscillating phase. The middle two red dots in Figure 1.(b) represents these configurations of the step sizes. The last column of Figure 2 shows another stable phase in Info-2. In this phase, $\tau/\tau$ is relatively large. The two time-scale dynamics are mixed, and another type of stable fixed points emerges.

**Phase diagram for $d = 2$.** Figure 3 shows the phase diagram when $d = 2$. In particular, the two red lines between Info-1 and Noninfo-1 in Figure 3 are determined by the left inequality in (15). In Info-1, both feature vectors are recovered by the generator. The dynamics of this phase are shown in Figure 1.(a) in the main text. In the Half-info phase, only the feature vector with the larger signal-to-noise ratio is recovered. The dynamics of this phase are shown in Figure 1.(c) in the main text. The blue line between Info-1 and oscillating phases shows the boundary between oscillation state and stable state.

Figure 1: (a): The locations of the five types of fixed points of the ODE (13). Their properties are listed in Table 1. (b): The phase diagram for the stationary state of the ODE (13). The colored lines illustrate the theoretical prediction of the boundaries between the different phases. Simulations results for a single numerical experiment are also shown to illustrate the oscillating phase: Each grey square represents the value of $\frac{1}{200}\int_{800}^{1000}[(P_t - \langle P_t \rangle)^2 + (q_t - \langle q_t \rangle)^2 + (r_t - \langle r_t \rangle)^2]\,\mathrm{d}t$ where $\langle P_t \rangle = \frac{1}{200}\int_{800}^{1000} P_t\,\mathrm{d}t$, and $\langle q_t \rangle$ and $\langle r_t \rangle$ are defined similarly. Note that the above quantity measures the variation (over time) of the training process as it approaches steady states. We see that the variation is indeed nonzero in the oscillating phase (see Figure 2), whereas the variation is close to zero in all other phases.

Figure 2: Macroscopic dynamics of Example 1 with $d = 1$. In the first row, the red, blue and yellow dots represent $P_t$, $q_t$, and $r_t$ respectively of the experimental results of a single trial. The black curves under the dots are theoretical predictions given by the ODE (13). We set a fix the discriminator's learning rate $\tau = 0.3$ and vary the generator's learning rate $\widetilde{\tau} = 0.03,\ 0.2,\ 0.4,\ 0.47$ from left to right column. These parameter settings are marked by the four red dots in the phase diagram in Figure 1. The second row is the phase portraits of the trajectories shown in the first row onto the $P$–$q$ plane. Figure (a) shows a case in the phase of info-1, where a subset of type (3) fixed points are stable. Figure (b) and (c) are in the oscillating phase, and (d) is in info-2, where the fixed points of type-5 are stable. The blue dots in the figures show the stable fixed points.

Figure 3: The phase diagram for the stationary states of the ODE (13) when $d = 2$. This phase diagram is generated by numerically computing the fixed points and eigenvalues of the Jacobian of the ODE (13).

## S-II  Heuristic derivations of the dynamics of the microscopic states

In this section, we derive the stochastic differential equations (10) in the main text for the microscopic states in a non-rigorous way. Specifically, we directly discard higher-order terms without any justification, in order to highlight the main ideas. In Section S-IV, we rigorously justify these steps by providing bounds on those terms.

Our starting point is the iterative algorithm (5) in the main text. Substituting the objective function $\mathcal{L}$ defined in (4) into (5), we have

$$\boldsymbol{w}_{k+1} = \boldsymbol{w}_k + \frac{\tau}{n}\left[\boldsymbol{y}_k f(\boldsymbol{y}_k^\top \boldsymbol{w}_k) - \widetilde{\boldsymbol{y}}_{2k}\widetilde{f}(\widetilde{\boldsymbol{y}}_{2k}^\top \boldsymbol{w}_k) - \lambda \boldsymbol{w}_k H'(\boldsymbol{w}_k^\top \boldsymbol{w}_k)\right] \tag{S-1}$$

$$\boldsymbol{V}_{k+1} = \boldsymbol{V}_k + \frac{\widetilde{\tau}}{n}\left[\boldsymbol{w}_k\widetilde{\boldsymbol{c}}_{2k+1}^\top \widetilde{f}(\widetilde{\boldsymbol{y}}_{2k+1}^\top \boldsymbol{w}_k) - \lambda \boldsymbol{V}_k \text{diag}(H'(\boldsymbol{V}_k^\top \boldsymbol{V}_k))\right], \tag{S-2}$$

where $\boldsymbol{y}_k$ and $\widetilde{\boldsymbol{y}}_k$ are true and fake samples generated according to (1) and (2) respectively. The two functions $f, \widetilde{f}$ stand for $f(x) = \frac{\mathrm{d}}{\mathrm{d}x}F(\widehat{D}(x))$ and $\widetilde{f}(x) = \frac{\mathrm{d}}{\mathrm{d}x}\widetilde{F}(\widehat{D}(x))$. The function $H'$ is derivative of $H$. If the input of $H'(\cdot)$ is a matrix, $H'$ applies to the input matrix element-wisely. The operation $\text{diag}(\boldsymbol{A})$ is a diagonal matrix of $\boldsymbol{A}$, where the off-diagonal term are set to zero.

We note that the elements of $\boldsymbol{w}_k$ and $\boldsymbol{V}_k$ are $\mathcal{O}(\frac{1}{\sqrt{n}})$ number as the norm of $\boldsymbol{w}_k$ and the norms of column vectors of $\boldsymbol{V}_k$ are all $\mathcal{O}(1)$ numbers. To investigate the dynamics of the microscopic state, it is convenient to rescale $\boldsymbol{w}_k$ and $\boldsymbol{V}_k$ by a factor of $\sqrt{n}$. We define $\widehat{\boldsymbol{u}}_i$ and $\widehat{\boldsymbol{v}}_{k,i}$ as the column view of the $i$'th row of the matrices $\sqrt{n}\boldsymbol{U}$ and $\sqrt{n}\boldsymbol{V}_k$ respectively, and $\widehat{w}_{k,i} \overset{\text{def}}{=} \sqrt{n}[\boldsymbol{w}_{k+1}]_i$. The update rule of $((\widehat{\boldsymbol{u}}_i, \widehat{\boldsymbol{v}}_{k,i}, \widehat{w}_{k,i})_{i=1,\dots,n})_{k=0,1,2,\dots}$ is

$$\widehat{w}_{k+1,i} - \widehat{w}_{k,i} = \frac{\tau}{n}\left[\left(\widehat{\boldsymbol{u}}_i^\top \boldsymbol{c}_k + \sqrt{n\eta_\mathrm{T}}a_{k,i}\right)f_k - \left(\widehat{\boldsymbol{v}}_{k,i}^\top \widetilde{\boldsymbol{c}}_{2k} + \sqrt{n\eta_\mathrm{G}}\widetilde{a}_{2k,i}\right)\widetilde{f}_{2k} - \lambda H'(z_k)\widehat{w}_{k,i}\right], \tag{S-3}$$

$$\widehat{\boldsymbol{v}}_{k+1,i} - \widehat{\boldsymbol{v}}_{k,i} = \frac{\widetilde{\tau}}{n}\left[\widehat{w}_{k,i}\widetilde{\boldsymbol{c}}_{2k+1}\widetilde{f}_{2k+1} - \lambda\text{diag}(H'(\boldsymbol{S}_k))\widehat{\boldsymbol{v}}_{k,i}\right], \tag{S-4}$$

where $a_{k,i}, \widetilde{a}_{k,i}$ are the $i$th elements of $\boldsymbol{a}_k$ and $\widetilde{\boldsymbol{a}}_k$ respectively, and $f_k$ and $\widetilde{f}_k$ are shorthands for

$$f_k = f(\boldsymbol{y}_k^\top \boldsymbol{w}_k/\sqrt{n}) = f\left(\boldsymbol{q}_k^\top \boldsymbol{c}_k + \sqrt{\frac{\eta_\mathrm{T}}{n}}\sum_{j=1}^n a_{k,j}\widehat{w}_{k,j}\right)$$

$$\widetilde{f}_k = \widetilde{f}(\widetilde{\boldsymbol{y}}_k^\top \boldsymbol{w}_{\lfloor k/2 \rfloor}/\sqrt{n}) = \widetilde{f}\left(\boldsymbol{r}_{\lfloor k/2 \rfloor}^\top \widetilde{\boldsymbol{c}}_k + \sqrt{\frac{\eta_\mathrm{G}}{n}}\sum_{j=1}^n \widetilde{a}_{k,j}\widehat{w}_{\lfloor k/2 \rfloor,j}\right),$$

respectively, and the empirical macroscopic quantities $\boldsymbol{q}_k, \boldsymbol{r}_k, \boldsymbol{z}_k$ and $\boldsymbol{S}_k$ are defined as follows

$$\boldsymbol{q}_k \overset{\text{def}}{=} \boldsymbol{U}^\top \boldsymbol{w}_k = \frac{1}{n}\sum_{i=1}^n \widehat{\boldsymbol{u}}_i\widehat{w}_i, \qquad \boldsymbol{r}_k \overset{\text{def}}{=} \boldsymbol{V}_k^\top \boldsymbol{w}_k = \frac{1}{n}\sum_{i=1}^n \widehat{\boldsymbol{v}}_{k,i}\widehat{w}_i,$$

$$z_k \overset{\text{def}}{=} \boldsymbol{w}_k^\top \boldsymbol{w}_k = \frac{1}{n}\sum_{i=1}^n \widehat{w}_{k,i}^2, \qquad \boldsymbol{S}_k \overset{\text{def}}{=} \boldsymbol{V}_k^\top \boldsymbol{V}_k = \frac{1}{n}\sum_{i=1}^n \widehat{\boldsymbol{v}}_{k,i}\widehat{\boldsymbol{v}}_{k,i}^\top, \tag{S-5}$$

$$\boldsymbol{P}_k \overset{\text{def}}{=} \boldsymbol{U}^\top \boldsymbol{V}_k = \frac{1}{n}\sum_{i=1}^n \widehat{\boldsymbol{u}}_i\widehat{\boldsymbol{v}}_{k,i}^\top.$$

The matrix $\boldsymbol{P}_k$ is not used in this section, but we put it here with the other macroscopic quantities for future reference.

Now we derive (10) from (S-3) and (S-4).

First, it is trivial to get the first equation of the SDE $\mathrm{d}\widehat{\boldsymbol{u}}_t = 0$ in (10) in the main text, since $\widehat{\boldsymbol{u}}_i$ does not change over time.

Next, we derive the second equation in (10). Averaging over $\widetilde{\boldsymbol{c}}_{2k+1}$ and $\widetilde{\boldsymbol{a}}_{2k+1}$ on the both sides of (S-4), we get

$$\left\langle \widehat{\boldsymbol{v}}_{k+1,i} - \widehat{\boldsymbol{v}}_{k,i} \right\rangle_{\widetilde{\boldsymbol{c}}_{2k+1}, \widetilde{\boldsymbol{a}}_{2k+1}}$$
$$= \frac{\widetilde{\tau}}{n}\left[ \left\langle \widetilde{f}\left( \boldsymbol{r}_k^\top \widetilde{\boldsymbol{c}} + \sqrt{\frac{\eta_{\mathrm{G}}}{n}}\sum_{j=1}^n [\widetilde{\boldsymbol{a}}]_j \widehat{w}_{k,j}\right)\widetilde{\boldsymbol{c}} \right\rangle_{\widetilde{\boldsymbol{c}},\widetilde{\boldsymbol{a}}} \widehat{w}_{k,i} - \lambda\mathrm{diag}(H'(\boldsymbol{S}_k))\widehat{\boldsymbol{v}}_{k,i} \right].$$

The bracket $\langle \cdot \rangle_{\widetilde{\boldsymbol{c}},\widetilde{\boldsymbol{a}}}$ here denotes the average over $\widetilde{\boldsymbol{c}} \sim \mathcal{P}_{\widetilde{\boldsymbol{c}}}$, and standard Gaussian vector $\widetilde{\boldsymbol{a}}$, where $\widetilde{\boldsymbol{c}}$ and $\widetilde{\boldsymbol{a}}$ are the random variables generating the fake sample in the generator as described in (2). Noting that $\widetilde{\boldsymbol{a}}$ is a Gaussian vector, the term $\frac{1}{\sqrt{n}}\sum_{j=1}^n [\widetilde{\boldsymbol{a}}]_j \widehat{w}_{k,j}$ in the above equation is also a Gaussian random variable, whose mean is zero and variance is $z_k$, which is defined in (S-5). Therefore, we have

$$\left\langle \widehat{\boldsymbol{v}}_{k+1,i} - \widehat{\boldsymbol{v}}_{k,i} \right\rangle_{\widetilde{\boldsymbol{c}}_{2k+1}, \widetilde{\boldsymbol{a}}_{2k+1}} = \frac{\widetilde{\tau}}{n}\left[ \widetilde{g}_k \widehat{w}_{k,i} + \boldsymbol{L}_k \widehat{\boldsymbol{v}}_{k,i} \right], \tag{S-6}$$

where

$$\widetilde{g}_k = \left\langle \widetilde{f}\left( \boldsymbol{r}_k^\top \widetilde{\boldsymbol{c}} + \sqrt{z_k \eta_{\mathrm{G}}} e \right]\right)\widetilde{\boldsymbol{c}} \right\rangle_{\widetilde{\boldsymbol{c}}, e} \tag{S-7}$$

$$\boldsymbol{L}_k = -\lambda\mathrm{diag}(H'(\boldsymbol{S}_k)), \tag{S-8}$$

where $\langle \cdot \rangle_{\widetilde{\boldsymbol{c}}, e}$ denotes the average over $\widetilde{\boldsymbol{c}} \sim \mathcal{P}_{\widetilde{\boldsymbol{c}}}$ and $e \sim \mathcal{N}(0,1)$. In addition, from (S-4), we also know that the second moment

$$\left\langle \left( \widehat{\boldsymbol{v}}_{k+1,i} - \widehat{\boldsymbol{v}}_{k,i}\right)^2 \right\rangle_{\widetilde{\boldsymbol{c}}_{2k+1}, \widetilde{\boldsymbol{a}}_{2k+1}} = \mathcal{O}(n^{-\frac{3}{2}}). \tag{S-9}$$

The moments estimations (S-6) and (S-9) imply the second equation in (10) in the main text. Since the second moments growth smaller than $\mathcal{O}(n^{-1})$, the differential equation for $\widehat{\boldsymbol{v}}_t$ has no diffusion term.

Finally, we derive the last equation in (10) in the main text from the update rule of $\widehat{w}_k$ (S-3). We observe that both the terms inside the function $f$ and outside of $f$ in (S-3) depend on $a_{k,i}$. Using Taylor's expansion, we linearize the contribution of $a_{k,i}$ to the function $f$:

$$f_k = f\left( \boldsymbol{q}_k^\top \boldsymbol{c}_k + \sqrt{\frac{\eta_{\mathrm{T}}}{n}}\sum_{j\neq i} a_{k,j}\widehat{w}_{k,j} + \sqrt{\frac{\eta_{\mathrm{T}}}{n}}a_{k,i}\widehat{w}_{k,i}\right)$$
$$= f(\boldsymbol{q}_k^\top \boldsymbol{c}_k + \sqrt{\tfrac{\eta_{\mathrm{T}}}{n}}\sum_{j\neq i} a_{k,j}\widehat{w}_{k,j}) + f'(\boldsymbol{q}_k^\top \boldsymbol{c}_k + \sqrt{\tfrac{\eta_{\mathrm{T}}}{n}}\sum_{j\neq i} a_{k,j}\widehat{w}_{k,j})\sqrt{\tfrac{\eta_{\mathrm{T}}}{n}}a_{k,i}\widehat{w}_{k,i} + \mathcal{O}(\tfrac{1}{n}). \tag{S-10}$$

Similarly, we have

$$\widetilde{f}_{2k} = \widetilde{f}(\boldsymbol{r}_k^\top \widetilde{\boldsymbol{c}}_{2k} + \sqrt{\tfrac{\eta_{\mathrm{G}}}{n}}\sum_{j\neq i}\widetilde{a}_{k,j}\widehat{w}_{k,j} + \sqrt{\tfrac{\eta_{\mathrm{G}}}{n}}\widetilde{a}_{2k,i}\widehat{w}_{k,i})$$
$$= \widetilde{f}(\boldsymbol{r}_k^\top \widetilde{\boldsymbol{c}}_{2k} + \sqrt{\tfrac{\eta_{\mathrm{G}}}{n}}\sum_{j\neq i}\widetilde{a}_{k,j}\widehat{w}_{k,j}) + \widetilde{f}'(\boldsymbol{r}_k^\top \widetilde{\boldsymbol{c}}_{2k} + \sqrt{\tfrac{\eta_{\mathrm{G}}}{n}}\sum_{j\neq i}\widetilde{a}_{k,j}\widehat{w}_{k,j})\sqrt{\tfrac{\eta_{\mathrm{G}}}{n}}\widetilde{a}_{2k,i}\widehat{w}_{k,i} + \mathcal{O}(\tfrac{1}{n}) \tag{S-11}$$

Substituting (S-10) and (S-11) into (S-3), we have

$$\frac{\widehat{w}_{k+1,i} - \widehat{w}_{k,i}}{\tau/n}$$

$$= \widehat{\boldsymbol{u}}_i^\top \boldsymbol{c}_k f(\boldsymbol{q}_k^\top \boldsymbol{c}_k + \sqrt{\tfrac{\eta_T}{n}}\textstyle\sum_{j\neq i} a_{k,j}\widehat{w}_{k,j}) - \widehat{\boldsymbol{v}}_{k,i}^\top \widetilde{\boldsymbol{c}}_{2k}\widetilde{f}(\boldsymbol{r}_k^\top \widetilde{\boldsymbol{c}}_{2k} + \sqrt{\tfrac{\eta_G}{n}}\textstyle\sum_{j\neq i}\widetilde{a}_{k,j}\widehat{w}_{k,j})$$

$$+ \widehat{w}_{k,i}\Big[a_{k,i}^2 f'(\boldsymbol{q}_k^\top \boldsymbol{c}_k + \sqrt{\tfrac{\eta_T}{n}}\textstyle\sum_{j\neq i}a_{k,j}\widehat{w}_{k,j}) - \widetilde{a}_{k,i}^2 \widetilde{f}'(\boldsymbol{r}_k^\top \widetilde{\boldsymbol{c}}_{2k} + \sqrt{\tfrac{\eta_G}{n}}\textstyle\sum_{j\neq i}\widetilde{a}_{2k,j}\widehat{w}_{k,j}) - \lambda H'(z_k)\Big]$$

$$+ \sqrt{n}\Big[a_{k,i}f(\boldsymbol{q}_k^\top \boldsymbol{c}_k + \sqrt{\tfrac{\eta_T}{n}}\textstyle\sum_{j\neq i}a_{k,j}\widehat{w}_{k,j}) + \widetilde{a}_{2k,i}\widetilde{f}'(\boldsymbol{r}_k^\top \widetilde{\boldsymbol{c}}_{2k} + \sqrt{\tfrac{\eta_G}{n}}\textstyle\sum_{j\neq i}\widetilde{a}_{2k,j}\widehat{w}_{k,j})\Big] + \delta_{k,i}, \tag{S-12}$$

where $\delta_{k,i}$ collects all higher-order terms whose contributions will vanish as $n \to \infty$. From this equation, we can already infer the SDE (10). Specifically, on the right hand side of (S-12), the terms in the first two lines correspond to the drift term in the SDE. Furthermore, the first term in the third line in (S-12) contributes to the SDE as a Brownian motion. More precisely, we can derive the third equation of the SDE (10) in the main text by the moments estimations. Specifically, the first-order moment is

$$\big\langle \widehat{w}_{k+1,i} - \widehat{w}_{k,i}\big\rangle_{\boldsymbol{c}_k,\boldsymbol{a}_k,\widetilde{\boldsymbol{c}}_{2k},\widetilde{\boldsymbol{a}}_{2k}} = \tfrac{\tau}{n}\Big[\widehat{\boldsymbol{u}}_i^\top \boldsymbol{g}_k - \widehat{\boldsymbol{v}}_{k,i}^\top \widetilde{\boldsymbol{g}}_k + \widehat{w}_{k,i}h_k\Big] + \mathcal{O}(n^{-\frac{3}{2}}) \tag{S-13}$$

where $\widetilde{\boldsymbol{g}}_k$ is defined in (S-7), and

$$\boldsymbol{g}_k = \Big\langle \boldsymbol{c}f(\boldsymbol{q}_k^\top \boldsymbol{c} + \sqrt{z_k\eta_T}e)\Big\rangle_{\boldsymbol{c},e} \tag{S-14}$$

$$h_k = \eta_T\Big\langle f'(\boldsymbol{q}_k^\top \boldsymbol{c} + \sqrt{z_k\eta_T}e)\Big\rangle_{\boldsymbol{c},e} - \widetilde{\eta}_G\Big\langle \widetilde{f}'(\boldsymbol{r}_k^\top \widetilde{\boldsymbol{c}} + \sqrt{z_k\eta_G}e)\Big\rangle_{\widetilde{\boldsymbol{c}},e} - \lambda H'(z_k). \tag{S-15}$$

The second moment is

$$\Big\langle \big(\widehat{w}_{k+1,i} - \widehat{w}_{k,i}\big)^2\Big\rangle_{\boldsymbol{c}_k,\boldsymbol{a}_k,\widetilde{\boldsymbol{c}}_{2k},\widetilde{\boldsymbol{a}}_{2k}} = \tfrac{\tau^2}{n}b_k + \mathcal{O}(n^{-\frac{3}{2}}), \tag{S-16}$$

where

$$b_k = \eta_T\Big\langle f^2(\boldsymbol{q}_k^\top \boldsymbol{c} + \sqrt{z_k\eta_T}e)\Big\rangle_{\boldsymbol{c},e} + \eta_G\Big\langle \widetilde{f}^2(\boldsymbol{r}_k^\top \widetilde{\boldsymbol{c}} + \sqrt{z_k\eta_G}e)\Big\rangle_{\widetilde{\boldsymbol{c}},e}. \tag{S-17}$$

From the (S-13) and (S-16), we derive the SDE for $\widehat{w}_t$ in (10) in the main text.

## S-III   Derive the ODE in Theorem 1 from the weak formulation of the PDE

In this section, we show how to derive the ODE (8) from the weak formulation of the PDE (12). Choosing the test function $\varphi$ being each element of $\widehat{\boldsymbol{u}}\widehat{\boldsymbol{v}}^\top$, $\widehat{\boldsymbol{u}}\widehat{w}$, $\widehat{\boldsymbol{v}}\widehat{w}$, $\widehat{\boldsymbol{v}}\widehat{\boldsymbol{v}}^\top$, $\widehat{w}^2$, and substituting those $\varphi$ into the weak formulation of the PDE (12), we will get the ODE (8) as presented in Theorem 1. In what follows, we provide additional details of this derivation.

We first derive the first ODE $\frac{\mathrm{d}}{\mathrm{d}t}\boldsymbol{P}_t = \dots$ in (8). Let $\varphi = [\widehat{\boldsymbol{u}}]_\ell[\widehat{\boldsymbol{v}}]_{\ell'}$, $\ell,\ell' = 1,2,\dots,d$, we have $\nabla_{\widehat{\boldsymbol{v}}}\varphi = [\widehat{\boldsymbol{u}}]_\ell \boldsymbol{s}_{\ell'}$, where $\boldsymbol{s}_{\ell'}$ is the $\ell'$th canonical basis (*i.e.*, all elements in $\boldsymbol{s}_{\ell'}$ are zeros, except that $\ell'$th element is 1). From the PDE (12) in the main text, we have $\forall \ell, \ell' = 1,2,\dots,d$:

$$\big\langle \mu_t, \varphi(\widehat{\boldsymbol{u}},\widehat{\boldsymbol{v}},\widehat{w})\big\rangle = \big\langle \mu_t, [\widehat{\boldsymbol{u}}]_\ell[\widehat{\boldsymbol{v}}]_{\ell'}\big\rangle = [\boldsymbol{P}_t]_{\ell,\ell'},$$

$$\big\langle \mu_t, (\widehat{w}\widetilde{\boldsymbol{g}}_t^\top + \widehat{\boldsymbol{v}}^\top \boldsymbol{L}_t)\nabla_{\widehat{\boldsymbol{v}}}\varphi\big\rangle = \big\langle \mu_t, ([\widehat{\boldsymbol{u}}]_\ell\widehat{w})[\widetilde{\boldsymbol{g}}_t]_{\ell'} + ([\widehat{\boldsymbol{u}}]_\ell\widehat{\boldsymbol{v}}^\top)[\boldsymbol{L}_t]_{:,\ell'}\big\rangle$$
$$= [\boldsymbol{q}_t]_l[\widetilde{\boldsymbol{g}}_t]_{\ell'} + [\boldsymbol{P}_t]_{\ell,:}[\boldsymbol{L}_t]_{:,\ell'},$$

where $[\boldsymbol{P}_t]_{\ell,:}$ and $[\boldsymbol{L}_t]_{:,\ell'}$ are $\ell$th row of $\boldsymbol{P}_t$ and $\ell'$th column of $\boldsymbol{L}$, respectively. In addition, we know that $\frac{\partial}{\partial \widehat{w}}\varphi = \frac{\partial^2}{\partial \widehat{w}^2}\varphi = 0$. Combining above results, we can recover the first ODE in (8).

Next, we derive the second ODE $\frac{d\boldsymbol{q}_t}{dt} = \dots$ in (8). Let $\varphi = [\widehat{\boldsymbol{u}}]_\ell\widehat{w}$, $\ell = 1,2,\dots,d$. We have $\nabla_{\widehat{\boldsymbol{v}}}\varphi = 0$, $\frac{\partial}{\partial \widehat{w}}\varphi = [\widehat{\boldsymbol{u}}]_\ell$ and $\frac{\partial^2}{\partial \widehat{w}^2}\varphi = 0$. Then $\forall \ell = 1,2,\dots,d$,

$$\big\langle \mu_t, \varphi(\widehat{\boldsymbol{u}},\widehat{\boldsymbol{v}},\widehat{w})\big\rangle = \big\langle \mu_t, [\widehat{\boldsymbol{u}}]_\ell\widehat{w}\big\rangle = [\boldsymbol{q}_t]_\ell$$

and

$$\left\langle \mu_t, (\widehat{\boldsymbol{u}}^\top \boldsymbol{g}_t - \widehat{\boldsymbol{v}}^\top \widetilde{\boldsymbol{g}}_t + h_t \widehat{w}) \frac{\partial}{\partial \widehat{w}} \varphi \right\rangle = \left\langle \mu_t, (\widehat{\boldsymbol{u}}^\top \boldsymbol{g}_t - \widehat{\boldsymbol{v}}^\top \widetilde{\boldsymbol{g}}_t + h_t \widehat{w})[\widehat{\boldsymbol{u}}]_\ell \right\rangle$$
$$= [\boldsymbol{g}_t]_\ell - [\boldsymbol{P}_t]_\ell \widetilde{\boldsymbol{g}}_t + [\boldsymbol{q}_t]_\ell h_t.$$

With above results, we can obtain the second ODE in (8).

Next, let's derive the ODE for $\frac{d\boldsymbol{S}_t}{dt}$. We set $\varphi = [\widehat{\boldsymbol{v}}]_\ell [\widehat{\boldsymbol{v}}]_{\ell'}$. If $\ell \neq \ell'$, we have $\nabla_{\widehat{v}} \varphi = [\widehat{\boldsymbol{v}}]_\ell \boldsymbol{s}_{\ell'} + [\widehat{\boldsymbol{v}}]_{\ell'} \boldsymbol{s}_\ell$, where $\boldsymbol{s}_{\ell'}$ is the $\ell'$th canonical basis. Then

$$\left\langle \mu_t, \varphi(\widehat{\boldsymbol{u}}, \widehat{\boldsymbol{v}}, \widehat{w}) \right\rangle = [\boldsymbol{S}_t]_{\ell,\ell'}$$

and

$$\left\langle \mu_t, (\widehat{w} \widetilde{\boldsymbol{g}}_t^\top + \widehat{\boldsymbol{v}}^\top \boldsymbol{L}_t) \nabla_{\widehat{v}} \varphi \right\rangle = \left\langle \mu_t, ([\widehat{\boldsymbol{v}}]_\ell \widehat{w})[\widetilde{\boldsymbol{g}}_t]_{\ell'} + ([\widehat{\boldsymbol{v}}]_\ell \widehat{\boldsymbol{v}}^\top)[\boldsymbol{L}_t]_{:,\ell'} \right\rangle$$
$$+ \left\langle \mu_t, ([\widehat{\boldsymbol{v}}]_{\ell'} \widehat{w})[\widetilde{\boldsymbol{g}}_t]_\ell + ([\widehat{\boldsymbol{v}}]_{\ell'} \widehat{\boldsymbol{v}}^\top)[\boldsymbol{L}_t]_{:,\ell} \right\rangle$$
$$= [\boldsymbol{r}_t]_\ell [\widetilde{\boldsymbol{g}}_t]_{\ell'} + [\widetilde{\boldsymbol{g}}_t]_\ell [\boldsymbol{r}_t]_{\ell'} + [\boldsymbol{S}_t]_{\ell,:}[\boldsymbol{L}_t]_{:,\ell'} + [\boldsymbol{L}_t]_{\ell,:}[\boldsymbol{S}_t]_{:,\ell'}$$

If $\ell = \ell'$, we have $\nabla_{\widehat{v}} \varphi = 2[\widehat{\boldsymbol{v}}]_\ell \boldsymbol{s}_\ell$, then

$$\left\langle \mu_t, \varphi(\widehat{\boldsymbol{u}}, \widehat{\boldsymbol{v}}, \widehat{w}) \right\rangle = [\boldsymbol{S}_t]_{\ell,\ell}$$

and

$$\left\langle \mu_t, (\widehat{w} \widetilde{\boldsymbol{g}}_t^\top + \widehat{\boldsymbol{v}}^\top \boldsymbol{L}_t) \nabla_{\widehat{v}} \varphi \right\rangle = 2([\boldsymbol{r}_t]_\ell [\widetilde{\boldsymbol{g}}_t]_\ell + [\boldsymbol{S}_t]_{\ell,:}[\boldsymbol{L}_t]_{:,\ell})$$

Plugging back the above two equations and combining the fact that $\frac{\partial}{\partial \widehat{w}} \varphi = \frac{\partial^2}{\partial \widehat{w}^2} \varphi = 0$, we recover the ODE of $\frac{d\boldsymbol{S}_t}{dt}$.

The rest two ODEs can be obtained in the similar way by letting $\varphi$ to be each distinct component of $\widehat{\boldsymbol{v}} \widehat{w}$ and $\widehat{w}^2$.

## S-IV    Proof of Theorem 1

In this section, we prove Theorem 1 shown in the main text. In the previous section, we have already provided a derivation of the ODE in Theorem 1 from the weak formulation of the PDE for the microscopic states. In this section, we follow a different path to prove the theorem without referencing the PDE, because it is easier to establish the rigorous bound of the convergence rate. Thus, the proof itself also provides another derivation of the ODE, where the most relevant part is Lemma 5.

### S-IV.1    Sketch of the proof

The proof follows the standard procedure of the convergence of stochastic processes [1, 2]. We here build the whole proof on Lemma 2 in the supplementary materials of [3]. For reader's convenient, we present that lemma below.

**Lemma 1** (Lemma 2 in the supplementary materials of [3]). *Consider a sequence of stochastic process* $\{\boldsymbol{x}_k^{(n)}, k = 0, 1, 2, \ldots, \lfloor nT \rfloor\}_{n=1,2,\ldots}$, *with some constant* $T > 0$. *If* $\boldsymbol{x}_k^{(n)}$ *can be decomposed into three parts*

$$\boldsymbol{x}_{k+1}^{(n)} - \boldsymbol{x}_k^{(n)} = \tfrac{1}{n} \phi(\boldsymbol{x}_k^{(n)}) + \boldsymbol{\rho}_k^{(n)} + \boldsymbol{\delta}_k^{(n)} \tag{S-18}$$

*such that*

*(C.1) The process* $\sum_{k'=0}^k \boldsymbol{\rho}_{k'}^{(n)}$ *is a martingale, and* $\mathbb{E} \|\boldsymbol{\rho}_k^{(n)}\|^2 \leq C(T)/n^{1+\epsilon_1}$ *for some positive* $\epsilon_1$;

*(C.2)* $\mathbb{E} \|\boldsymbol{\delta}_k^{(n)}\| \leq C(T)/n^{1+\epsilon_2}$ *for some positive* $\epsilon_2$;

*(C.3)* $\phi(\boldsymbol{x})$ *is a Lipschitz function, i.e.,* $\|\phi(\boldsymbol{x}) - \phi(\widetilde{\boldsymbol{x}})\| \leq C\|\boldsymbol{x} - \widetilde{\boldsymbol{x}}\|$;

*(C.4)* $\mathbb{E} \|\boldsymbol{x}_k^{(n)}\|^2 \leq C$ *for all* $k \leq \lfloor nT \rfloor$;

*(C.5)* $\mathbb{E} \|\boldsymbol{x}_0^{(n)} - \boldsymbol{x}_0^*\| \leq C/n^{\epsilon_3}$ *for some positive* $\epsilon_3$ *and a deterministic vector* $\boldsymbol{x}_0^*$,

*then we have*

$$\|\boldsymbol{x}_k^{(n)} - \boldsymbol{x}(\tfrac{k}{n})\| \le C(T)n^{-\min\{\frac{1}{2}\epsilon_1, \epsilon_2, \epsilon_3\}},$$

*where $\boldsymbol{x}(t)$ is the solution of the ODE*

$$\tfrac{\mathrm{d}}{\mathrm{d}t}\boldsymbol{x}(t) = \phi(\boldsymbol{x}(t)), \quad \text{with } \boldsymbol{x}(0) = \boldsymbol{x}_0^*.$$

In Theorem 1, the stochastic process is the macroscopic states $\{\boldsymbol{M}_k, k = 0, 1, \ldots\}$, where $\boldsymbol{M}_k$ is a symmetric matrix consists of 5 non-trivial parts $\boldsymbol{P}_k$, $\boldsymbol{q}_k$, $\boldsymbol{r}_k$, $\boldsymbol{S}_k$, and $z_k$ as shown in (6) in the main text. Following (S-18), we have the following decomposition for $\boldsymbol{M}_k$

$$\boldsymbol{M}_{k+1} - \boldsymbol{M}_k = \tfrac{1}{n}\phi(\boldsymbol{M}_k) + (\boldsymbol{M}_{k+1} - \mathbb{E}_k\,\boldsymbol{M}_{k+1}) + [\mathbb{E}_k\,\boldsymbol{M}_{k+1} - \boldsymbol{M}_k - \tfrac{1}{n}\phi(\boldsymbol{M}_k)], \quad \text{(S-19)}$$

in which the matrix-valued function $\phi(\boldsymbol{M})$ represents the functions on the right hand sides of the ODE (8), and $\mathbb{E}_k$ denotes the conditional expectation given the state of the Markov chain $\boldsymbol{X}_k$. Note that the stochastic process of the macroscopic state $\boldsymbol{M}_k$ is driven by the Markov chain of the microscopic state $\boldsymbol{X}_k$. Thus, $\mathbb{E}_k$ is well-defined. For future reference, we denotes $\mathbb{E}$ the unconditional expectation of all the randomness of the Markov chain $\boldsymbol{X}_k$, *i.e.*, the initial state $\boldsymbol{U}, \boldsymbol{V}_0, \boldsymbol{w}_0$ and $\{\boldsymbol{a}_k, \boldsymbol{c}_k, \widetilde{\boldsymbol{a}}_k, \widetilde{\boldsymbol{c}}_k | k = 0, 1, 2, \ldots\}$. By definition, $\sum_{k'=0}^k (\boldsymbol{M}_{k'+1} - \mathbb{E}_{k'}\boldsymbol{M}_{k'})$ is a Martingale.

### S-IV.2  Check the conditions provided in Lemma 1

In this subsection, we check the condition (C.1)–(C.5) for the decomposition of (S-19). Once all conditions are proved to be satisfied, Theorem 1 will be proved.

We first note that (C.5) is the assumption (A.5) in the main text. Thus, (C.5) is satisfied. Before proving other conditions, we declare a lemma.

**Lemma 2.** *Under the same setting as Theorem 1, given $T > 0$, then*

$$\mathbb{E}\left(\sum_{\ell=1}^d [\boldsymbol{V}_k]_{i,\ell}^4 + [\boldsymbol{w}_k]_i^4\right) \le C(T)n^{-2}, \quad \forall i = 1, 2, \ldots, n, \text{ and } k = 0, 1, \ldots, \lfloor nT \rfloor, \quad \text{(S-20)}$$

The proof can be founded in Section S-IV.3.

### Check Condition (C.4)

**Lemma 3.** *Under the same setting as Theorem 1, for all $k = 0, 1, \ldots, \lfloor nT \rfloor$ with a given $T > 0$, then*

$$\mathbb{E}\|\boldsymbol{P}_k\|^2 \le C(T), \qquad\qquad\qquad \mathbb{E}\|\boldsymbol{q}_k\|^2 \le C(T),$$
$$\mathbb{E}\|\boldsymbol{S}_k\|^2 \le C(T), \qquad\qquad\qquad \mathbb{E}z_k^2 \le C(T),$$
$$\mathbb{E}\|\boldsymbol{r}_k\|^2 \le C(T).$$

*Proof.* It's a direct consequence of Lemma 2. We first verify $\mathbb{E}z_k^2 \le C(T)$. Using Holder's inequality, we have

$$\mathbb{E}z_k^2 = \mathbb{E}\left(\sum_{i=1}^n w_{k,i}^2\right)^2 \le n\mathbb{E}\sum_{i=1}^n w_{k,i}^4 \le C(T)$$

For $[\boldsymbol{S}_k]_{\ell,\ell}$, $\ell = 1, \ldots, d$, similarly, we have

$$\mathbb{E}[\boldsymbol{S}_k]_{\ell,\ell}^2 = \mathbb{E}\left(\sum_{i=1}^n [\boldsymbol{V}_k]_{i,\ell}^2\right)^2 \le C(T).$$

and for $\mathbb{E}[\boldsymbol{S}_k]_{\ell,\ell'}^2$, $\ell \ne \ell'$, we have:

$$\mathbb{E}[\boldsymbol{S}_k]_{\ell,\ell'}^2 = \mathbb{E}\left(\sum_{i=1}^n [\boldsymbol{V}_k]_{i,\ell}[\boldsymbol{V}_k]_{i,\ell'}\right)^2$$
$$\le \mathbb{E}\left(\sum_{i=1}^n [\boldsymbol{V}_k]_{i,\ell}^2\right)\left(\sum_{i=1}^n [\boldsymbol{V}_k]_{i,\ell'}^2\right)$$
$$\le \sqrt{\mathbb{E}\left(\sum_{i=1}^n [\boldsymbol{V}_k]_{i,\ell}^2\right)^2 \mathbb{E}\left(\sum_{i=1}^n [\boldsymbol{V}_k]_{i,\ell'}^2\right)^2}$$
$$\le C(T)$$

where in reaching the third and last line, we used the Cauchy-Schwartz inequality. Now, we get $\mathbb{E}\|\boldsymbol{S}_k\|^2 \le C(T)$. The rest bounds of $\mathbb{E}\|\boldsymbol{P}_k\|^2$, $\mathbb{E}\|\boldsymbol{q}_k\|^2$ and $\mathbb{E}\|\boldsymbol{r}_k\|^2$ in Lemma 3 can also be directly verified using the Cauchy-Schwartz inequality. $\qquad\square$

**Check Condition (C.3)**

**Lemma 4.** *If Assumption (A.3) hold, $\phi(\boldsymbol{M})$ is a Lipschitz function.*

*Proof.* It suffices to verify each component of gradient $\nabla\phi(\boldsymbol{M})$ is bounded. Assumption (A.3) ensures that $H'$ is Lipschitz and the derivatives up to fourth order of the functions $f$, $\widetilde{f}$ exists and uniformly bounded. These conditions guarantee that the partial derivatives of $\phi(\boldsymbol{M})$ w.r.t. $\boldsymbol{P}$, $\boldsymbol{q}$, $\boldsymbol{S}$ and $\boldsymbol{r}$ are bounded. The remaining thing is to show that $\frac{\partial\phi(\boldsymbol{M})}{\partial z}$ is also bounded. Since there is a $\sqrt{z}$ term in $\phi(\boldsymbol{M})$, the boundness can be potentially broken at $z = 0$. However, we can show that it is not the case. For example, we can show that $\langle \boldsymbol{c}f(\boldsymbol{c}^\top\boldsymbol{q} + e\sqrt{z})\rangle_{\boldsymbol{c},e}$ is a Lipschitz function, because

$$
\begin{aligned}
\tfrac{\partial}{\partial z}\langle \boldsymbol{c}f(\boldsymbol{c}^\top\boldsymbol{q} + e\sqrt{z})\rangle_{\boldsymbol{c},e} &= \tfrac{1}{2}z^{-\frac{1}{2}}\langle ecf'(cq + e\sqrt{z})\rangle_{\boldsymbol{c},e} \\
&= \tfrac{1}{2}\langle cf''(cq + e\sqrt{z})\rangle_{\boldsymbol{c},e}
\end{aligned}
$$

is always a well-defined bounded function. In reaching the first line, we here interchanged the expectation and derivative, which is valid because of the boundness of $f(\cdot)$, and in reaching the second line, we used the Stein's lemma. Finally, other terms in (9) involving $\sqrt{z}$ can be treated in the same way. Thus, $\phi(\boldsymbol{M})$ is a Lipschitz function. $\qquad\square$

**Check Condition (C.2)**

**Lemma 5.** *Under the same setting as Theorem 1, for all $k = 0, 1, \ldots, \lfloor nT\rfloor$ with a given $T > 0$, then*

$$
\mathbb{E}\left\|\mathbb{E}_k\,\boldsymbol{M}_{k+1} - \boldsymbol{M}_k - \tfrac{1}{n}\phi(\boldsymbol{M}_k)\right\| \le C(T)n^{-\frac{3}{2}}.
$$

*Proof.* The above inequality can be split into 5 parts

$$
\mathbb{E}\left\|\mathbb{E}_k\,\boldsymbol{P}_{k+1} - \boldsymbol{P}_k - \tfrac{\widetilde{\tau}}{n}(\boldsymbol{q}_k\widetilde{\boldsymbol{g}}_k^\top + \boldsymbol{P}_k\boldsymbol{L}_k)\right\| \le C(T)n^{-\frac{3}{2}} \quad\text{(S-21)}
$$

$$
\mathbb{E}\left\|\mathbb{E}_k\,\boldsymbol{q}_{k+1} - \boldsymbol{q}_k - \tfrac{\tau}{n}\left(\boldsymbol{g}_k - \boldsymbol{P}_k\widetilde{\boldsymbol{g}}_k + \boldsymbol{q}_k h_k\right)\right\| \le C(T)n^{-\frac{3}{2}} \quad\text{(S-22)}
$$

$$
\mathbb{E}\left\|\mathbb{E}_k\,\boldsymbol{S}_{k+1} - \boldsymbol{S}_k - \tfrac{\widetilde{\tau}}{n}\left(\boldsymbol{r}_k\widetilde{\boldsymbol{g}}_k^\top + \widetilde{\boldsymbol{g}}_k\boldsymbol{r}_k^\top + \boldsymbol{S}_k\boldsymbol{L}_k + \boldsymbol{L}_k\boldsymbol{S}_k\right)\right\| \le C(T)n^{-\frac{3}{2}} \quad\text{(S-23)}
$$

$$
\mathbb{E}\left\|\mathbb{E}_k\,z_{k+1} - z_k - \tfrac{2\tau}{n}\left(\boldsymbol{q}_k^\top\boldsymbol{g}_k - \boldsymbol{r}_k^\top\widetilde{\boldsymbol{g}}_k + z_k h_k\right) - \tfrac{\tau^2}{n}b_k\right\| \le C(T)n^{-\frac{3}{2}}, \quad\text{(S-24)}
$$

$$
\mathbb{E}\left\|\mathbb{E}_k\,\boldsymbol{r}_{k+1} - \boldsymbol{r}_k - \tfrac{\tau}{n}\left(\boldsymbol{P}_k^\top\boldsymbol{g}_k - \boldsymbol{S}_k\widetilde{\boldsymbol{g}}_k + \boldsymbol{r}_k h_k\right) - \tfrac{\widetilde{\tau}}{n}\left(z_k\widetilde{\boldsymbol{g}}_k + \boldsymbol{L}_k\boldsymbol{r}_k\right)\right\| \le C(T)n^{-\frac{3}{2}} \quad\text{(S-25)}
$$

where $\widetilde{\boldsymbol{g}}_k$, $\boldsymbol{L}_k$, $\boldsymbol{g}_k$, $h_k$, $b_k$ are defined in (S-7), (S-8), (S-14), (S-15) and (S-17), respectively.

We first prove (S-21). From (S-2), we have

$$
\boldsymbol{V}_{k+1} - \boldsymbol{V}_k = \tfrac{\widetilde{\tau}}{n}\left[\boldsymbol{w}_k\widetilde{\boldsymbol{c}}_{2k+1}^\top\widetilde{f}(\widetilde{\boldsymbol{c}}_{2k+1}^\top\boldsymbol{V}_k^\top\boldsymbol{w}_k + \eta_{\mathrm{G}}\widetilde{\boldsymbol{a}}_{2k+1}^\top\boldsymbol{w}_k) - \lambda\boldsymbol{V}_k\mathrm{diag}(H'(\boldsymbol{S}_k))\right]. \quad\text{(S-26)}
$$

Averaging both sides of the above equation over $\widetilde{\boldsymbol{c}}_{2k+1}$ and $\widetilde{\boldsymbol{a}}_{2k+1}$, we have

$$
\mathbb{E}_k\,\boldsymbol{V}_{k+1} - \boldsymbol{V}_k = \tfrac{\widetilde{\tau}}{n}\left[\boldsymbol{w}_k\widetilde{\boldsymbol{g}}_k^\top + \boldsymbol{V}_k\boldsymbol{L}_k\right], \quad\text{(S-27)}
$$

where $\widetilde{\boldsymbol{g}}_k$ and $\boldsymbol{L}_k$ are defined in (S-7) and (S-8), respectively. Multiplying $\boldsymbol{U}^\top$ from the left on the both sides of the above equation, we have

$$
\mathbb{E}_k\,\boldsymbol{P}_{k+1} - \boldsymbol{P}_k = \tfrac{\widetilde{\tau}}{n}\left[\boldsymbol{q}_k\widetilde{\boldsymbol{g}}_k^\top + \boldsymbol{P}_k\boldsymbol{L}_k\right],
$$

which implies (S-21). In fact, there is no higher-order term in (S-21), and the left hand side of (S-21) is exactly zero.

Then, we prove (S-22). From (S-1), we have

$$
\boldsymbol{w}_{k+1} - \boldsymbol{w}_k = \tfrac{\tau}{n}\left[\boldsymbol{y}_k f(\boldsymbol{y}_k^\top\boldsymbol{w}_k) - \widetilde{\boldsymbol{y}}_{2k}\widetilde{f}(\widetilde{\boldsymbol{y}}_{2k}^\top\boldsymbol{w}_k) - \lambda\boldsymbol{w}_k\mathrm{diag}(H'(z_k))\right], \quad\text{(S-28)}
$$

where $\boldsymbol{y}_k = \boldsymbol{U}\boldsymbol{c}_k + \sqrt{\eta_{\mathrm{T}}}\boldsymbol{a}_k$ and $\widetilde{\boldsymbol{y}}_{2k} = \boldsymbol{V}_k\widetilde{\boldsymbol{c}}_{2k} + \sqrt{\eta_{\mathrm{G}}}\widetilde{\boldsymbol{a}}_{2k}$. Averaging both sides of the above equation over $\boldsymbol{c}_k$, $\boldsymbol{a}_k\widetilde{\boldsymbol{c}}_{2k}$ and $\widetilde{\boldsymbol{a}}_{2k}$, we have

$$
\begin{aligned}
\mathbb{E}_k\,\boldsymbol{w}_{k+1} - \boldsymbol{w}_k = \tfrac{\tau}{n}\Big[ & \boldsymbol{U}\boldsymbol{g}_k + \left\langle \boldsymbol{a}_k f(\boldsymbol{c}_k^\top \boldsymbol{q}_k + \sqrt{\eta_{\mathrm{T}}}\boldsymbol{a}_k^\top \boldsymbol{w}_k) \right\rangle \\
& - \boldsymbol{V}_k\widetilde{\boldsymbol{g}}_k - \left\langle \widetilde{\boldsymbol{a}}_{2k}\widetilde{f}(\widetilde{\boldsymbol{c}}_{2k}^\top \boldsymbol{r}_k + \sqrt{\eta_{\mathrm{G}}}\widetilde{\boldsymbol{a}}_{2k}^\top \boldsymbol{w}_k) \right\rangle - \lambda\boldsymbol{w}_k\mathrm{diag}(H'(z_k))\Big].
\end{aligned}
$$

Multiplying $\boldsymbol{U}^\top$ from the left on the both sides of the above equation, we have

$$
\begin{aligned}
\mathbb{E}_k\,\boldsymbol{q}_{k+1} - \boldsymbol{q}_k = \tfrac{\tau}{n}\Big[ & \boldsymbol{g}_k - \boldsymbol{P}_k\widetilde{\boldsymbol{g}}_k + \sqrt{\eta_{\mathrm{T}}}\left\langle \boldsymbol{U}^\top \boldsymbol{a}_k f(\boldsymbol{c}_k^\top \boldsymbol{q}_k + \sqrt{\eta_{\mathrm{T}}}\boldsymbol{a}_k^\top \boldsymbol{w}_k) \right\rangle_{\boldsymbol{c},\boldsymbol{a}} \\
& - \sqrt{\eta_{\mathrm{G}}}\left\langle \boldsymbol{U}^\top \widetilde{\boldsymbol{a}}\widetilde{f}(\widetilde{\boldsymbol{c}}^\top \boldsymbol{r}_k + \sqrt{\eta_{\mathrm{G}}}\widetilde{\boldsymbol{a}}^\top \boldsymbol{w}_k) \right\rangle_{\widetilde{\boldsymbol{c}},\widetilde{\boldsymbol{a}}} - \lambda\boldsymbol{q}_k\mathrm{diag}(H'(z_k))\Big] \qquad \text{(S-29)}
\end{aligned}
$$

We note that $\begin{bmatrix} \boldsymbol{U}^\top \boldsymbol{a}_k \\ \boldsymbol{w}_k^\top \boldsymbol{a}_k \end{bmatrix}$ are Gaussian random vector with zero-mean and covariance matrix $\begin{bmatrix} \boldsymbol{I} & \boldsymbol{q}_k \\ \boldsymbol{q}_k^\top & z_k \end{bmatrix}$.
We can rewrite

$$
\begin{aligned}
\left\langle \boldsymbol{U}^\top \boldsymbol{a} f(\boldsymbol{c}^\top \boldsymbol{q}_k + \sqrt{\eta_{\mathrm{T}}}\boldsymbol{a}^\top \boldsymbol{w}_k) \right\rangle_{\boldsymbol{c},\boldsymbol{a}} &= z_k^{-1/2}\boldsymbol{U}^\top \boldsymbol{w}_k \left\langle e f(\boldsymbol{c}^\top \boldsymbol{q}_k + \sqrt{z_k\eta_{\mathrm{T}}}e) \right\rangle_{\boldsymbol{c},e} \qquad \text{(S-30)} \\
&= \sqrt{\eta_T}\boldsymbol{q}_k \left\langle f'(\boldsymbol{c}^\top \boldsymbol{q}_k + \sqrt{z_k\eta_{\mathrm{T}}}e) \right\rangle_{\boldsymbol{c},e},
\end{aligned}
$$

where the second line is due to Stein's lemma (i.e., integral by part for Gaussian random variable.)
Similarly, we have

$$
\left\langle \boldsymbol{U}^\top \widetilde{\boldsymbol{a}}\widetilde{f}(\widetilde{\boldsymbol{c}}^\top \boldsymbol{r}_k + \sqrt{\eta_{\mathrm{G}}}\widetilde{\boldsymbol{a}}^\top \boldsymbol{w}_k) \right\rangle_{\widetilde{\boldsymbol{c}},\widetilde{\boldsymbol{a}}} = \sqrt{\eta_{\mathrm{G}}}\boldsymbol{q}_k \left\langle \widetilde{f}'(\widetilde{\boldsymbol{c}}^\top \boldsymbol{r}_k + \sqrt{z_k\eta_{\mathrm{G}}}e) \right\rangle_{\widetilde{\boldsymbol{c}},e}. \qquad \text{(S-31)}
$$

Substituting (S-30) and (S-31) into (S-29), we get

$$
\mathbb{E}_k\,\boldsymbol{q}_{k+1} - \boldsymbol{q}_k = \tfrac{\tau}{n}\left[\boldsymbol{g}_k - \boldsymbol{P}_k\widetilde{\boldsymbol{g}}_k + \boldsymbol{q}_k h_k\right],
$$

where $\widetilde{\boldsymbol{g}}_k$, $\boldsymbol{g}_k$, and $h_k$ are defined in (S-7), (S-14), and (S-15), respectively. Now, we proved (S-22), which again has no higher-order term.

We next prove (S-23). Note that

$$
\begin{aligned}
\boldsymbol{S}_{k+1} - \boldsymbol{S}_k &= (\boldsymbol{V}_k + \boldsymbol{V}_{k+1} - \boldsymbol{V}_k)^\top(\boldsymbol{V}_k + \boldsymbol{V}_{k+1} - \boldsymbol{V}_k) - \boldsymbol{S}_k \\
&= \boldsymbol{V}_k^\top(\boldsymbol{V}_{k+1} - \boldsymbol{V}_k) + (\boldsymbol{V}_{k+1} - \boldsymbol{V}_k)^\top\boldsymbol{V}_k + (\boldsymbol{V}_{k+1} - \boldsymbol{V}_k)^\top(\boldsymbol{V}_{k+1} - \boldsymbol{V}_k).
\end{aligned}
$$

Averaging both sides of the above equation over $\widetilde{\boldsymbol{c}}_{2k+1}$ and $\widetilde{\boldsymbol{a}}_{2k+1}$ and substituting (S-27) into above equation, we have

$$
\mathbb{E}_k\,\boldsymbol{S}_{k+1} - \boldsymbol{S}_k = \tfrac{\widetilde{\tau}}{n}\left[\boldsymbol{r}_k\widetilde{\boldsymbol{g}}_k^\top + \boldsymbol{S}_k\boldsymbol{L}_k + \widetilde{\boldsymbol{g}}_k\boldsymbol{r}_k^\top + \boldsymbol{L}_k\boldsymbol{S}_k\right] + \tfrac{\widetilde{\tau}^2}{n^2}\left[\boldsymbol{w}_k\widetilde{\boldsymbol{g}}_k^\top + \boldsymbol{V}_k\boldsymbol{L}_k\right]^\top\left[\boldsymbol{w}_k\widetilde{\boldsymbol{g}}_k^\top + \boldsymbol{V}_k\boldsymbol{L}_k\right].
$$

$$\text{(S-32)}$$

We know that

$$
\begin{aligned}
\mathbb{E}\,\|\left[\boldsymbol{w}_k\widetilde{\boldsymbol{g}}_k^\top + \boldsymbol{V}_k\boldsymbol{L}_k\right]^\top\left[\boldsymbol{w}_k\widetilde{\boldsymbol{g}}_k^\top + \boldsymbol{V}_k\boldsymbol{L}_k\right]\| &\le \mathbb{E}\,\|\boldsymbol{w}_k\widetilde{\boldsymbol{g}}_k^\top + \boldsymbol{V}_k\boldsymbol{L}_k\|^2 \\
&\le 2z_k\|\widetilde{\boldsymbol{g}}_k\|^2 + 2\|\boldsymbol{S}_k\|\|\boldsymbol{L}_k\|^2 \\
&\le C\mathbb{E}\left[z_k + \|\boldsymbol{S}_k\|\right] \\
&\le C(T), \qquad \text{(S-33)}
\end{aligned}
$$

where $\widetilde{\boldsymbol{g}}_k$, $\boldsymbol{L}_k$ are defined in (S-7) and (S-8), respectively. The third line of the above inequalities is due to the fact that $\widetilde{f}$ and $H'$ are uniformly bounded, and in reaching the last line, we used Lemma 3. Combining (S-32) and (S-33), we reach (S-23).

The other two inequalities (S-24) and (S-25) can be proved in a similar way. We omit the details here. $\qquad \square$

**Check Condition (C.1)**

**Lemma 6.** *Under the same setting as Theorem 1, for all $k = 0, 1, \ldots, \lfloor nT \rfloor$ with a given $T > 0$, then*

$$\mathbb{E} \left\| \boldsymbol{M}_{k+1} - \mathbb{E}_k \, \boldsymbol{M}_{k+1} \right\|^2 \leq C(T) n^{-2}.$$

*Proof.* Note that $\mathbb{E} \left\| \boldsymbol{M}_{k+1} - \mathbb{E}_k \, \boldsymbol{M}_{k+1} \right\|^2 = \mathbb{E} \left\| \boldsymbol{M}_{k+1} - \boldsymbol{M}_k - \mathbb{E}_k \left( \boldsymbol{M}_{k+1} - \boldsymbol{M}_k \right) \right\|^2 \leq \mathbb{E} \left\| \boldsymbol{M}_{k+1} - \boldsymbol{M}_k \right\|^2$. It is sufficient to prove

$$\mathbb{E} \left\| \boldsymbol{M}_{k+1} - \boldsymbol{M}_k \right\|^2 \leq C(T) n^{-2}. \tag{S-34}$$

In what follows, we are going to bound the second-order moment of each element in $\boldsymbol{M}_{k+1} - \boldsymbol{M}_k$. In particular, we bound the 5 blocks $\boldsymbol{P}_k$, $\boldsymbol{S}_k$, $\boldsymbol{q}_k$, $z_k$ and $\boldsymbol{r}_k$ of $\boldsymbol{M}_k$ separately.

We first bound $\mathbb{E} \left\| \boldsymbol{P}_{k+1} - \boldsymbol{P}_k \right\|^2$. Multiplying $\boldsymbol{U}^\top$ from left on both sides of (S-26), we have

$$\boldsymbol{P}_{k+1} - \boldsymbol{P}_k = \tfrac{\widetilde{\tau}}{n} \left[ \boldsymbol{q}_k \widetilde{\boldsymbol{c}}_{2k+1}^\top \widetilde{f}(\widetilde{\boldsymbol{c}}_{2k+1}^\top \boldsymbol{V}_k^\top \boldsymbol{w}_k + \eta_{\mathrm{G}} \widetilde{\boldsymbol{a}}_{2k+1}^\top \boldsymbol{w}_k) - \lambda \boldsymbol{P}_k \mathrm{diag}(H'(\boldsymbol{V}_k^\top \boldsymbol{V}_k)) \right]$$

We then get

$$\mathbb{E} \left\| \boldsymbol{P}_{k+1} - \boldsymbol{P}_k \right\|^2 \leq C n^{-2} \mathbb{E} \left[ \|\boldsymbol{q}_k\|^2 \mathbb{E}_k \|\widetilde{\boldsymbol{c}}_{2k+1}\|^2 + \|\boldsymbol{P}_k\|^2 \right]$$

$$\leq C n^{-2} \mathbb{E} \left[ 1 + \|\boldsymbol{q}_k\|^2 + \|\boldsymbol{P}_k\|^2 \right]$$

$$\leq C(T) n^{-2}. \tag{S-35}$$

Here the last line is due to Lemma 3.

We next bound $\mathbb{E} \left\| \boldsymbol{q}_{k+1} - \boldsymbol{q}_k \right\|^2$ in the same way. Specifically, multiplying $\boldsymbol{U}^\top$ from the left on both sides of (S-28), we get

$$\boldsymbol{q}_{k+1} - \boldsymbol{q}_k = \tfrac{\tau}{n} \left[ \boldsymbol{U}^\top \boldsymbol{y}_k f(\boldsymbol{y}_k^\top \boldsymbol{w}_k) - \boldsymbol{U}^\top \widetilde{\boldsymbol{y}}_{2k} \widetilde{f}(\widetilde{\boldsymbol{y}}_{2k}^\top \boldsymbol{w}_k) - \lambda \boldsymbol{q}_k \mathrm{diag}(H'(\boldsymbol{w}_k^\top \boldsymbol{w}_k)) \right].$$

We then have

$$\mathbb{E} \left\| \boldsymbol{q}_{k+1} - \boldsymbol{q}_k \right\|^2$$

$$\leq \tfrac{\tau^2}{n^2} \mathbb{E} \left[ \|\boldsymbol{c}_k\|^2 f_k^2 + \|\boldsymbol{U}^\top \boldsymbol{a}_k\|^2 f_k^2 + \|\boldsymbol{P}_k\|^2 \|\widetilde{\boldsymbol{c}}_{2k}\|^2 \widetilde{f}_{2k}^2 + \|\boldsymbol{U}^\top \widetilde{\boldsymbol{a}}_{2k}\|^2 \widetilde{f}_{2k}^2 + \|\boldsymbol{q}_k\|^2 h_k^2 \right]$$

$$\leq C n^{-2} \left[ 1 + \sqrt{\mathbb{E} \|\boldsymbol{U}^\top \boldsymbol{a}_k\|^4} \sqrt{\mathbb{E} f_k^4} + \sqrt{\mathbb{E} \|\boldsymbol{U}^\top \widetilde{\boldsymbol{a}}_{2k}\|^4} \sqrt{\mathbb{E} \widetilde{f}_{2k}^4} + \mathbb{E} z_k^2 + \mathbb{E} \|\boldsymbol{S}_k\|^2 \right]$$

$$\leq C n^{-2} [1 + \mathbb{E} z_k^2 + \mathbb{E} \|\boldsymbol{S}_k\|^2]$$

$$\leq C(T) n^{-2}, \tag{S-36}$$

where $f_k$ and $\widetilde{f}_{2k}$ are shorthands for $f(\boldsymbol{y}_k^\top \boldsymbol{w}_k)$ and $\widetilde{f}(\widetilde{\boldsymbol{y}}_{2k}^\top \boldsymbol{w}_k)$ respectively. In reaching the last line, we used Lemma 3 again.

Similarly, we can also prove that

$$\mathbb{E} \left\| \boldsymbol{S}_{k+1} - \boldsymbol{S}_k \right\|^2 \leq C(T) n^{-2}$$

$$\mathbb{E} \left( z_{k+1} - z_k \right)^2 \leq C(T) n^{-2} \tag{S-37}$$

$$\mathbb{E} \left\| \boldsymbol{r}_{k+1} - \boldsymbol{r}_k \right\|^2 \leq C(T) n^{-2}.$$

Combining (S-35), (S-36) and (S-37), we can prove (S-34), which concludes the whole proof. $\quad\square$

## S-IV.3  Proof of Lemma 2

Before proving Lemma 2, we first present and prove the following lemma. Let $\boldsymbol{u}_i$ and $\boldsymbol{v}_{k,i}$ denote the $i$th row vectors of $\boldsymbol{U}$ and $\boldsymbol{V}_k$ in column view, respectively, and let $w_{k,i}$ be the $i$th element of the vector $\boldsymbol{w}_k$.

**Lemma 7.** *Under the same setting as Theorem 1, for all $k = 0, 1, \ldots, \lfloor nT \rfloor$ with a given $T > 0$, then*

$$\left\| \mathbb{E}_k \, \boldsymbol{v}_{k+1,i} - \boldsymbol{v}_{k,i} \right\| \leq C n^{-1} \left( \|\boldsymbol{v}_{k,i}\| + |w_{k,i}| \right) \tag{S-38}$$

$$\left| \mathbb{E}_k \, w_{k,i} - w_{k,i} \right| \leq C n^{-1} \left( \|\boldsymbol{u}_i\| + \|\boldsymbol{v}_{k,i}\| + |w_{k,i}| \right). \tag{S-39}$$

In the proof of this lemma and Lemma 2, we omit the two constants $\eta_{\mathrm{T}}$ and $\eta_{\mathrm{G}}$ for simplicity.

*Proof.* From (S-2) and knowing that the function $\widetilde{f}$ and $H'$ are uniformly bounded, we can immediately prove (S-38).

Next, we are going to prove (S-39). From (S-1), we know

$$
\begin{aligned}
&\left| \mathbb{E}_k\, w_{k+1,i} - w_{k,i} \right| \\
&\leq \tfrac{\tau}{n} \left( \left| \boldsymbol{u}_i^\top \left\langle \boldsymbol{c}_k f(\boldsymbol{y}_k^\top \boldsymbol{w}_k) \right\rangle_{\boldsymbol{c}_k,\boldsymbol{a}_k} \right| + \left| \left\langle a_{k,i} f(\boldsymbol{y}_k^\top \boldsymbol{w}_k) \right\rangle_{\boldsymbol{c}_k,\boldsymbol{a}_k} \right| \right. \\
&\qquad \left. + \left| \boldsymbol{v}_{k,i}^\top \left\langle \widetilde{\boldsymbol{c}}_{2k} \widetilde{f}(\widetilde{\boldsymbol{y}}_{2k}^\top \boldsymbol{w}_k) \right\rangle_{\widetilde{\boldsymbol{c}}_{2k},\widetilde{\boldsymbol{a}}_{2k}} \right| + \left| \left\langle \widetilde{a}_{2k,i} \widetilde{f}(\widetilde{\boldsymbol{y}}_{2k}^\top \boldsymbol{w}_k) \right\rangle_{\widetilde{\boldsymbol{c}}_{2k},\widetilde{\boldsymbol{a}}_{2k}} \right| + \lambda \left| w_{k,i} H'(\boldsymbol{w}_k^\top \boldsymbol{w}_k) \right| \right) \\
&\leq C n^{-1} \left( \|\boldsymbol{u}_i\| + \|\boldsymbol{v}_{k,i}\| + |w_{k,i}| + \left| \left\langle a_{k,i} f(\boldsymbol{y}_k^\top \boldsymbol{w}_k) \right\rangle_{\boldsymbol{c}_k,\boldsymbol{a}_k} \right| + \left| \left\langle \widetilde{a}_{2k,i} \widetilde{f}(\widetilde{\boldsymbol{y}}_{2k}^\top \boldsymbol{w}_k) \right\rangle_{\widetilde{\boldsymbol{c}}_{2k},\widetilde{\boldsymbol{a}}_{2k}} \right| \right),
\end{aligned}
\tag{S-40}
$$

where the last is due to the fact that $H'$, $f$ and $\widetilde{f}$ are uniformly bounded. Using Taylor's expansion up-to zero-order

$$
\begin{aligned}
f(\boldsymbol{y}_k^\top \boldsymbol{w}_k) &= f(\boldsymbol{q}_k^\top \boldsymbol{c}_k + \textstyle\sum_{j\neq i} w_{k,j} a_{k,j} + w_{k,j} a_{k,j}) \\
&= f(\boldsymbol{q}_k^\top \boldsymbol{c}_k + \textstyle\sum_{j\neq i} w_{k,j} a_{k,j}) + f'(\boldsymbol{q}_k^\top \boldsymbol{c}_k + \textstyle\sum_{j\neq i} w_{k,j} a_{k,j} + \chi_{k,i}) w_{k,j} a_{k,j},
\end{aligned}
$$

with $\chi_{k,i}$ being some number such that $|\chi_{k,i}| \leq |w_{k,i} a_{k,i}|$, we have

$$
\begin{aligned}
&\left| \left\langle a_{k,i} f(\boldsymbol{y}_k^\top \boldsymbol{w}_k) \right\rangle_{\boldsymbol{c}_k,\boldsymbol{a}_k} \right| \\
&\leq \left| \left\langle f(\boldsymbol{q}_k^\top \boldsymbol{c}_k + \textstyle\sum_{j\neq i} w_{k,j} a_{k,j}) a_{k,i} \right\rangle_{\boldsymbol{c}_k,\boldsymbol{a}_k} \right| + \left| \left\langle f'(\boldsymbol{q}_k^\top \boldsymbol{c}_k + \textstyle\sum_{j\neq i} w_{k,j} a_{k,j} + \chi_{k,i}) w_{k,j} a_{k,j}^2 \right\rangle_{\boldsymbol{c}_k,\boldsymbol{a}_k} \right| \\
&= \left| \left\langle f'(\boldsymbol{q}_k^\top \boldsymbol{c}_k + \textstyle\sum_{j\neq i} w_{k,j} a_{k,j} + \chi_{k,i}) w_{k,i} a_{k,i}^2 \right\rangle_{\boldsymbol{c}_k,\boldsymbol{a}_k} \right| \\
&\leq C |w_{k,i}|.
\end{aligned}
\tag{S-41}
$$

The second line is due to the fact $a_{k,i}$ is zero-mean, and in reaching the last line, we used the boundness of $f'$. Similarly, we can get

$$
\left| \left\langle \widetilde{a}_{2k,i} \widetilde{f}(\widetilde{\boldsymbol{y}}_{2k}^\top \boldsymbol{w}_k) \right\rangle_{\widetilde{\boldsymbol{c}}_{2k},\widetilde{\boldsymbol{a}}_{2k}} \right| \leq C |w_{k,i}|.
\tag{S-42}
$$

Substituting (S-41) and (S-42) into (S-40), we prove (S-40). $\qquad\square$

Now we are in the position to prove Lemma 2.

*Proof of Lemma 2.* Because of the exchangeability, $\mathbb{E}\, w_{k,i}^4 = \mathbb{E}\, w_{k,j}^4$, and $\mathbb{E}\,[\boldsymbol{V}_k]_{i,\ell}^4 = \mathbb{E}\,[\boldsymbol{V}_k]_{j,\ell}^4$ for all $i,j = 1, 2, \ldots, n$ and $\ell = 1, 2, \ldots, d$. Thus, we only need to prove (S-20) for any specific $i$.

We first prove $\mathbb{E}\, w_{k,i}^4 \leq C(T) n^{-2}$. We know that

$$
\mathbb{E}\, w_{k+1,i}^4 - \mathbb{E}\, w_{k,i}^4 = 4\mathbb{E}\left[ w_{k,i}^3 \mathbb{E}_k\left( w_{k+1,i} - w_{k,i} \right) \right] + 6\mathbb{E}\left[ w_{k,i}^2 \mathbb{E}_k\left( w_{k+1,i} - w_{k,i} \right)^2 \right]
\tag{S-43}
$$

$$
+ 4\mathbb{E}\left[ w_{k,i} \mathbb{E}_k\left( w_{k+1,i} - w_{k,i} \right)^3 \right] + \mathbb{E}\mathbb{E}_k\left( w_{k+1,i} - w_{k,i} \right)^4.
$$

From (S-1) and knowing that $h$, $f$ and $\widetilde{f}$ are uniformly bounded, we have

$$
\mathbb{E}_k\left( w_{k+1,i} - w_{k,i} \right)^\gamma \leq \frac{C}{n^\gamma} \left( 1 + \|\boldsymbol{u}_i\|^\gamma + \|\boldsymbol{v}_{k,i}\|^\gamma + |w_{k,i}|^\gamma \right) \quad \text{for } \gamma = 2, 3, 4.
\tag{S-44}
$$

Substituting (S-39) and (S-44) into (S-43) and using the Young's inequality, we have

$$
\begin{aligned}
\mathbb{E}\, w_{k+1,i}^4 - \mathbb{E}\, w_{k,i}^4 &\leq \tfrac{C}{n} \left( n^{-2} + \mathbb{E}\,\|\boldsymbol{u}_i\|^4 + \mathbb{E}\,\|\boldsymbol{v}_{k,i}\|^4 + \mathbb{E}\, w_{k,i}^4 \right). \\
&\leq \tfrac{C}{n} \mathbb{E}\left( n^{-2} + \textstyle\sum_{\ell=1}^d [\boldsymbol{V}_k]_{i,\ell}^4 + w_{k,i}^4 \right),
\end{aligned}
\tag{S-45}
$$

where the last line is due to Assumption A.4), which implies $\sum_\ell [\boldsymbol{U}]_{i,\ell}^4 \le C$. Similarly, we can prove

$$\sum_{\ell=1}^d \mathbb{E}\left([\boldsymbol{V}_{k+1}]_{i,\ell}^4 - [\boldsymbol{V}_k]_{i,\ell}^4\right) \le \frac{C}{n}\mathbb{E}\left(n^{-2} + \sum_{\ell=1}^d [\boldsymbol{V}_k]_{i,\ell}^4 + w_{k,i}^4\right). \quad (\text{S-46})$$

Combining (S-45) and (S-46), we have

$$\mathbb{E}\left(w_{k+1,i}^4 + \sum_{\ell=1}^d [\boldsymbol{V}_{k+1}]_{i,\ell}^4\right) - \mathbb{E}\left(w_{k,i}^4 + \sum_{\ell=1}^d [\boldsymbol{V}_k]_{i,\ell}^4\right) \le \frac{C}{n}\left[n^{-2} + \mathbb{E}\left(w_{k,i}^4 + \sum_{\ell=1}^d [\boldsymbol{V}_k]_{i,\ell}^4\right)\right].$$

Using the above inequality iteratively, we have

$$\mathbb{E}\left(w_{k,i}^4 + \sum_{\ell=1}^d [\boldsymbol{V}_k]_{i,\ell}^4\right) \le \left(n^{-2} + w_{0,i}^4 + \sum_{\ell=1}^d [\boldsymbol{V}_0]_{i,\ell}^4\right)e^{\frac{k}{n}C}.$$

Since $\mathbb{E}\left(w_{0,i}^4 + \sum_{\ell=1}^d [\boldsymbol{V}_0]_{i,\ell}^4\right)$ are bounded in Assumption A.4), we now reach (S-20). $\qquad\square$

## S-V  Local stability analysis of the fixed points of the ODE

In this section, we provide additional details on the local stability analysis of the ODE for Example 1. We first its simplified ODE (13) in the main text. Then, we provide the derivation of the local stability analysis when $d = 1$, where the main results are summarized in Section S-I. Finally, we establish the proof of Claim 1 in the main text.

### S-V.1  Derive the reduced ODE for Example 1 when $\lambda \to \infty$

In Example 1, $f(x) = \widetilde{f}(x) = x$. Plugging back to (9), we obtain that

$$\begin{aligned}
\boldsymbol{g}_t &= \boldsymbol{\Lambda}\boldsymbol{q}_t \\
\widetilde{\boldsymbol{g}}_t &= \widetilde{\boldsymbol{\Lambda}}\boldsymbol{r}_t \\
b_t &= \eta_{\mathrm{T}}(\boldsymbol{q}_t^\top \boldsymbol{\Lambda}\boldsymbol{q}_t + \eta_{\mathrm{T}}z_t) + \eta_{\mathrm{G}}(\boldsymbol{r}_t^\top \widetilde{\boldsymbol{\Lambda}}\boldsymbol{r}_t + \eta_{\mathrm{G}}z_t).
\end{aligned} \quad (\text{S-47})$$

Correspondingly, ODE in (8) becomes:

$$\begin{aligned}
\tfrac{\mathrm{d}}{\mathrm{d}t}\boldsymbol{P}_t &= \widetilde{\tau}\big(\boldsymbol{q}_t\widetilde{\boldsymbol{r}}_t^\top \widetilde{\boldsymbol{\Lambda}} + \boldsymbol{P}_t\boldsymbol{L}_t\big) \\
\tfrac{\mathrm{d}}{\mathrm{d}t}\boldsymbol{q}_t &= \tau\big(\boldsymbol{\Lambda}\boldsymbol{q}_t - \boldsymbol{P}_t\widetilde{\boldsymbol{\Lambda}}\boldsymbol{r}_t + \boldsymbol{q}_t h_t\big) \\
\tfrac{\mathrm{d}}{\mathrm{d}t}\boldsymbol{r}_t &= \tau\big(\boldsymbol{P}_t^T \boldsymbol{\Lambda}\boldsymbol{q}_t - \boldsymbol{S}_t\widetilde{\boldsymbol{\Lambda}}\boldsymbol{r}_t + \boldsymbol{r}_t h_t\big) + \widetilde{\tau}\big(\widetilde{\boldsymbol{\Lambda}}\boldsymbol{r}_t + \boldsymbol{L}_t\boldsymbol{r}_t\big) \\
\tfrac{\mathrm{d}}{\mathrm{d}t}\boldsymbol{S}_t &= \widetilde{\tau}\big(\boldsymbol{r}_t\boldsymbol{r}_t^\top \widetilde{\boldsymbol{\Lambda}}^\top + \widetilde{\boldsymbol{\Lambda}}\boldsymbol{r}_t\boldsymbol{r}_t^\top + \boldsymbol{S}_t\boldsymbol{L}_t + \boldsymbol{L}_t\boldsymbol{S}_t\big) \\
\tfrac{\mathrm{d}}{\mathrm{d}t}z_t &= 2\tau(\boldsymbol{q}_t^\top \boldsymbol{\Lambda}\boldsymbol{q}_t - \boldsymbol{r}_t^\top \widetilde{\boldsymbol{\Lambda}}\boldsymbol{r}_t + z_t h_t) \\
&\quad + \tau^2[\eta_{\mathrm{T}}(\boldsymbol{q}_t^\top \boldsymbol{\Lambda}\boldsymbol{q}_t + z_t\eta_{\mathrm{T}}) + \eta_{\mathrm{G}}(\boldsymbol{r}_t^\top \widetilde{\boldsymbol{\Lambda}}\boldsymbol{r}_t + z_t\eta_{\mathrm{G}})]
\end{aligned} \quad (\text{S-48})$$

The first four equations are exactly (13). From last two equations of (S-48), by setting $\tfrac{\mathrm{d}}{\mathrm{d}t}\mathrm{diag}\{\boldsymbol{S}_t\} = \boldsymbol{0}$, $\tfrac{\mathrm{d}}{\mathrm{d}t}z_t = 0$, $\mathrm{diag}(\boldsymbol{S}_t) = \boldsymbol{I}$ and $z_t = 1$, we can get (14).

### S-V.2  A complete study of all fixed points when $d = 1$

We next provide the local stability analysis of the fixed points of the ODE (13). When $d = 1$ and $\lambda \to \infty$, the macroscopic state is described by only 3 scalars, $P_t$, $q_t$ and $r_t$. The result is summarized in Table 1. For the sake of simplicity, we only consider the case $\Lambda = \widetilde{\Lambda}$, and set $\eta_{\mathrm{T}} = \eta_{\mathrm{G}} = 1$, but all analysis can be extended to general cases.

The fixed points are given by the condition $\tfrac{d}{dt}P_t = \tfrac{d}{dt}q_t = \tfrac{d}{dt}r_t = 0$. From (13), we get

$$\begin{cases}
\widetilde{\tau}\Lambda r\,(q - rP) = 0 \\
\tau\big[\Lambda - \tau - \Lambda\big(1 + \tfrac{\tau}{2}\big)q^2\big]q - \tau\Lambda\big[P + \big(\tfrac{\tau}{2} - 1\big)rq\big]r = 0 \\
\tau\Lambda Pq + \big[\Lambda(\widetilde{\tau} - \tau) - \tau^2\big]r + \Lambda\big(\tau - \widetilde{\tau} - \tfrac{\tau^2}{2}\big)r^3 - \tau\Lambda\big(1 + \tfrac{\tau}{2}\big)rq^2 = 0,
\end{cases} \quad (\text{S-49})$$

where $P, q, r$ are the stationary macroscopic state. The local stability of a fixed point is identified by whether the Jacobian matrix

$$J(P, q, r) \overset{\text{def}}{=} \begin{bmatrix} \frac{\partial}{\partial P} g_1 & \frac{\partial}{\partial q} g_1 & \frac{\partial}{\partial r} g_1 \\ \frac{\partial}{\partial P} g_3 & \frac{\partial}{\partial q} g_3 & \frac{\partial}{\partial r} g_3 \\ \frac{\partial}{\partial P} g_5 & \frac{\partial}{\partial q} g_5 & \frac{\partial}{\partial r} g_5 \end{bmatrix}$$

has eigenvalue with non-negative real part or not, where $g_1 = \tilde{\tau}\Lambda r (q - rP)$, $g_2 = \tau \left[\Lambda - \tau - \Lambda \left(1 + \frac{\tau}{2}\right) q^2\right] q - \tau\Lambda \left[P + \left(\frac{\tau}{2} - 1\right) rq\right] r$ and $g_5 = \tau\Lambda Pq + \left[\Lambda(\tilde{\tau}-\tau) - \frac{(\eta_{\mathrm{T}}+\eta_{\mathrm{G}})\tau^2}{2}\right] r + \Lambda \left(\tau - \tilde{\tau} - \frac{\tau^2 \eta_{\mathrm{G}}}{2}\right) r^3 - \tau\Lambda \left(1 + \frac{\tau \eta_{\mathrm{T}}}{2}\right) rq^2$.

**Type (1) fixed point at $P = q = r = 0$**

It is easy to verify that $q = r = 0$ and any $P \in [-1, 1]$ is a solution of (S-49), but we first consider $P = 0$.

The Jacobian at $P = q = r = 0$ is

$$J(0, 0, 0) = \begin{bmatrix} 0 & 0 & 0 \\ 0 & \tau(\Lambda - \tau) & 0 \\ 0 & 0 & \Lambda(\tilde{\tau} - \tau) - \tau^2 \end{bmatrix}.$$

Thus, type (1) fixed point is stable if and only if

$$\tau \geq \Lambda \quad \text{and} \quad \frac{\tilde{\tau}}{\tau} \leq \frac{\tau + \Lambda}{\Lambda}.$$

**Type (2) fixed points at $P = q = 0$, $r = \pm r^* \neq 0$**

We first analyze when such fixed point exists and then study its local stability.

If $P = q = 0$, the first two equations in (S-49) trivially hold. The third equation becomes

$$\tau[\Lambda(r^2 - 1) - \frac{\tau}{2}(\Lambda r^2 + 2)] - \tilde{\tau}\Lambda(r^2 - 1) = 0.$$

The solution is

$$r^2 = \frac{\tau - \tilde{\tau} + \tau^2/\Lambda}{\tau - \tilde{\tau} - \tau^2/2}. \tag{S-50}$$

Since only the positive solution corresponds a fixed one. Thus, type (2) fixed point exists if

$$\frac{\tilde{\tau}}{\tau} \leq 1 - \frac{\tau}{2} \tag{S-51}$$

$$\text{or} \quad \frac{\tilde{\tau}}{\tau} \geq \frac{\tau + \Lambda}{\Lambda}. \tag{S-52}$$

Next, we investigate the local stability of this fixed point. The Jacobian at $\tilde{q} = q = 0$ for a given $r$ is

$$J(0, 0, r) = \begin{bmatrix} -\tilde{\tau}\Lambda r^2 & \tilde{\tau}\Lambda r & 0 \\ -\tau\Lambda r & \tau(\Lambda - \tau) - \Lambda\tau(\frac{\tau}{2} - 1)r^2 & 0 \\ 0 & 0 & 3r^2\Lambda(\tau - \frac{\tau^2}{2} - \tilde{\tau}) - \tau^2 + \Lambda(\tilde{\tau} - \tau) \end{bmatrix}$$

$$\tag{S-53}$$

Plugging (S-50) into $[J(0, 0, r)]_{3,3}$ of (S-53), then $[J(0, 0, r)]_{3,3} \leq 0$ implies

$$\frac{\tilde{\tau}}{\tau} \geq \frac{\tau}{\Lambda} + 1.$$

It indicates that the stationary points at the region (S-51) are always unstable. Thus, we only need to consider the second region specified by (S-52).

For the upper-left $2 \times 2$ sub-matrix of (S-53), the eigenvalues are non-positive if and only if

$$-\tilde{\tau}\Lambda r^2 + \tau(\Lambda - \tau) - \Lambda\tau(\frac{\tau}{2} - 1)r^2 \leq 0 \tag{S-54}$$

$$\tau + \Lambda(\frac{\tau}{2} - 1)r^2 + \Lambda - \Lambda \geq 0. \tag{S-55}$$

Plugging (S-50) into (S-54), we can get

$$\frac{\tilde{\tau}}{\tau} \geq 2. \tag{S-56}$$

Plugging (S-50) into (S-55) and combining (S-52), we can get

$$[\tau + \Lambda(\tfrac{\tau}{2} - 1)]\tilde{\tau} \geq \tau\Lambda(\tfrac{\tau}{2} - 1).$$

Solving this inequality implies that

$$\frac{\tilde{\tau}}{\tau} \leq \frac{(\tfrac{\tau}{2} - 1)\Lambda}{(\tfrac{\tau}{2} - 1)\Lambda + \tau}, \quad \text{when } \tau < \frac{2\Lambda}{\Lambda + 2} \tag{S-57}$$

and

$$\frac{\tilde{\tau}}{\tau} \geq \frac{(\tfrac{\tau}{2} - 1)\Lambda}{(\tfrac{\tau}{2} - 1)\Lambda + \tau}, \quad \text{when } \tau > \frac{2\Lambda}{\Lambda + 2}. \tag{S-58}$$

Note that (S-58) is included by (S-56), as $\frac{(\frac{\tau}{2}-1)\Lambda}{(\frac{\tau}{2}-1)\Lambda+\tau} \leq 2$ when $\tau > \frac{2\Lambda}{\Lambda+2}$.

Then, combining (S-52), (S-56), and (S-57) we obtain the stability region for $\tilde{q} = q = 0$,

$$\frac{\tilde{\tau}}{\tau} \geq 1 + \frac{\tau}{\Lambda}, \; \frac{\tilde{\tau}}{\tau} \geq 2, \; \text{and } \frac{\tilde{\tau}}{\tau} \leq \beta(\tau),$$

where $\beta(\tau)$ is defined as

$$\beta(\tau) \overset{\text{def}}{=} \begin{cases} \frac{(\frac{\tau}{2}-1)\Lambda}{(\frac{\tau}{2}-1)\Lambda+\tau} & \text{if } \tau \leq \frac{2\Lambda}{\Lambda+2} \\ +\infty & \text{otherwise.} \end{cases}$$

**Type (3) fixed points at $q = r = 0$ and $|P| \in (0, 1]$**

As mentioned, we can check that $q = r = 0$ and any $P \in [-1, 1]$ is a solution of (S-49). We next investigate the stable region for the fixed point $P = \pm 1$ and $q = r = 0$, which represents the perfect recovery state. For general $P$, we can analyze its fixed point similarly.

The Jacobian at $q = r = 0$ for any given $P$ is

$$J(1, 0, 0) = \begin{bmatrix} 0 & 0 & 0 \\ 0 & \tau(\Lambda - \tau) & -\tau\Lambda \\ 0 & \tau\Lambda & \Lambda(\tilde{\tau} - \tau) - \tau^2 \end{bmatrix}.$$

In this case, $J(1, 0, 0)$ always has an eigenvalue $0$ and to calculate the rest two eigenvalues, we only need to analyze the bottom-right $2 \times 2$ sub-matrix of $J(\tilde{q})$. The characteristic polynomial of this sub-matrix is $f(\lambda) = \lambda^2 - (a + d)\lambda + ad - bc$, where $a = \tau(\Lambda - \tau)$, $b = -\tau\Lambda$, $c = \tau\Lambda$, and $d = \Lambda(\tilde{\tau} - \tau) - \tau^2$. The roots of $f(\lambda) = 0$ both have non-positive real part if and only if $a + d \leq 0$, $ad - bc \geq 0$, which implies

$$\frac{\tilde{\tau}}{\tau} \leq \frac{2\tau}{\Lambda} \quad \text{and} \quad \frac{\tilde{\tau}}{\tau}(\tau - \Lambda) \leq \frac{\tau^2}{\Lambda}. \tag{S-59}$$

Noting that when $\tau < \Lambda$, the second inequality always hold, and when $\tau > \Lambda$, $\frac{\tau^2}{\Lambda(\tau-\Lambda)} \geq 4$, we can combine the two inequalities in (S-59) into compact form

$$\frac{\tilde{\tau}}{\tau} \leq \min\{\tfrac{2\tau}{\Lambda}, \max\{\tfrac{\tau^2}{\Lambda|\tau-\Lambda|}, 4\}\}.$$

The stable regions of the fixed points for $q = r = 0$ and $|P| < 1$ can be derived in a similar way, which turns out to be a subset of the stable region for $P = \pm 1$.

**Type (4) fixed point at $P = r = 0$ and $q \neq 0$.**

From (S-49), we know when at fixed point, $\tilde{q} = r = 0$, then $q^2 = \frac{\Lambda - \tau}{\Lambda(1 + \tau/2)}$, so $\tau$ must satisfy $\tau \leq \Lambda$. The corresponding Jacobian is:

$$J(0, 0, q) = \begin{bmatrix} 0 & 0 & \tilde{\tau}\Lambda q \\ 0 & \tau(\Lambda - \tau) - 3\tau\Lambda q^2(1 + \tfrac{\tau}{2}) & 0 \\ \tau\Lambda q & 0 & (\tilde{\tau} - \tau)\Lambda - \tau^2 - \tau\Lambda q^2(1 + \tfrac{\tau}{2}) \end{bmatrix}.$$

After plugging in $q^2 = \frac{\Lambda - \tau}{\Lambda(1 + \tau/2)}$, we can obtain that the characteristic function $\det(\lambda \boldsymbol{I} - J(0,0,q))$ is equal to:

$$\det(\lambda \boldsymbol{I} - J(0,0,q)) = [\lambda + 2\tau(\Lambda - \tau)][\lambda(\lambda + (2\tau - \widetilde{\tau})\Lambda) - \tau\widetilde{\tau}\Lambda^2 q^2]$$

Clearly, $\det(\lambda \boldsymbol{I} - J(0,0,q)) = 0$ has a non-negative root, so $J(0,0,q)$ always has a non-negative eigenvalue. This means type (4) fixed points are always unstable.

**Type (5) fixed points at $P, q, r \neq 0$**

The fixed points equation (S-49) can also have solutions that none of $P$, $q$ and $r$ is zero. In what follows, we derive the analytical expression of this type of solutions. It turns out that there can be maximum 8 solutions, which are symmetric by flipping the signs. We are unable to derive the analytical expression for their stable region, but it can be computed numerically.

If $P, q, r \neq 0$, (S-49) yields

$$r = \frac{q}{P} \tag{S-60}$$

$$\Lambda - \tau - \Lambda(1 + \tfrac{\tau}{2})q^2 - \Lambda[\tfrac{P}{q} + (\tfrac{\tau}{2} - 1)r]r = 0 \tag{S-61}$$

$$\tau\Lambda\widetilde{P}q + r\left[\Lambda(\widetilde{\tau} - \tau) - \tau^2\right] + r^3\Lambda\left(\tau - \widetilde{\tau} - \tfrac{\tau^2}{2}\right) - rq^2\tau\Lambda\left(1 + \tfrac{\tau}{2}\right) = 0. \tag{S-62}$$

Plugging (S-60) into (S-61), we can get

$$q^{-2} = -\tfrac{1}{\tau}[\Lambda(\tfrac{\tau}{2} - 1)P^{-2} + \Lambda(1 + \tfrac{\tau}{2})]. \tag{S-63}$$

Then combining (S-60) (S-63) and (S-62), we can obtain the following equations:

$$AP^{-4} + BP^{-2} + C = 0 \tag{S-64}$$

where $A = \Lambda(\widetilde{\tau} - \tau)(\tfrac{1}{2} - \tfrac{1}{\tau}) + \widetilde{\tau}$, $B = \Lambda[\tfrac{\widetilde{\tau}}{\tau}(1 + \tfrac{\tau}{2}) - 2]$, $C = \Lambda(1 + \tfrac{\tau}{2})$. We can find that (S-64) is an equation of $P^{-2}$ with at most two roots. Combining (S-63), we know there are at most 2 solutions for the pair $(q^{-2}, P^{-2})$ and hence there are at most 8 solutions for $(q, P, r)$, where $r = P/q$.

## S-V.3 Proof of Claim 1

*Proof of Claim 1.* We first compute the Jacobian $\partial\{\frac{\mathrm{d}}{\mathrm{d}t}\boldsymbol{P}_t, \frac{\mathrm{d}}{\mathrm{d}t}\boldsymbol{q}_t, \frac{\mathrm{d}}{\mathrm{d}t}\boldsymbol{r}_t, \}/\partial\{\boldsymbol{P}_t, \boldsymbol{q}_t, \boldsymbol{r}_t\}$ of the ODE (13) when $\boldsymbol{q}_t = \boldsymbol{r}_t = \boldsymbol{0}$. In the Jacobian, the $d \times d$ matrix $\boldsymbol{P}_t$ is considered as a $d^2$ vector. In fact, all elements in the Jacobian matrix related to $\boldsymbol{P}_t$ are 0. Specifically, the Jacobian for any $\boldsymbol{P}$ and $\boldsymbol{q}_t = \boldsymbol{r}_t = \boldsymbol{0}$ is

$$\boldsymbol{J}(\boldsymbol{P}) = \begin{bmatrix} \boldsymbol{0} & \boldsymbol{0} & \boldsymbol{0} \\ \boldsymbol{0} & \tau(\boldsymbol{\Lambda} - \tau\overline{\eta^2}\boldsymbol{I}_d) & -\tau\boldsymbol{P}\widetilde{\boldsymbol{\Lambda}} \\ \boldsymbol{0} & \tau\boldsymbol{P}^\top\boldsymbol{\Lambda} & \widetilde{\boldsymbol{\Lambda}}(\widetilde{\tau} - \tau) - \tau^2\overline{\eta^2} \end{bmatrix}, \tag{S-65}$$

where $\overline{\eta^2} = (\eta_T^2 + \eta_G^2)/2$.

When $\boldsymbol{P}$ is diagonal, under a suitable column-row permutation, the $\boldsymbol{J}(\boldsymbol{P})$ in (S-65) becomes a block diagonal matrix, where each non-zero block is a $2 \times 2$ matrix

$$\begin{bmatrix} \tau([\boldsymbol{\Lambda}]_{\ell,\ell} - \tau\overline{\eta^2}) & -\tau[\boldsymbol{P}]_{\ell,\ell}[\widetilde{\boldsymbol{\Lambda}}]_{\ell,\ell} \\ \tau[\boldsymbol{P}]_{\ell,\ell}[\boldsymbol{\Lambda}]_{\ell,\ell} & [\widetilde{\boldsymbol{\Lambda}}]_{\ell,\ell}(\widetilde{\tau} - \tau) - \tau^2\overline{\eta^2} \end{bmatrix} \tag{S-66}$$

for $\ell = 1, 2, \ldots, d$. Intuitively, the above matrix is the Jacobian matrix of $\partial\{\frac{\mathrm{d}}{\mathrm{d}t}[\boldsymbol{q}_t]_\ell, \frac{\mathrm{d}}{\mathrm{d}t}[\boldsymbol{r}_t]_\ell\}/\partial\{[\boldsymbol{q}_t]_\ell, [\boldsymbol{r}_t]_\ell\}$, and the Jacobian $\partial\{\frac{\mathrm{d}}{\mathrm{d}t}[\boldsymbol{q}_t]_\ell, \frac{\mathrm{d}}{\mathrm{d}t}[\boldsymbol{r}_t]_\ell\}/\partial\{[\boldsymbol{q}_t]_{\ell'}, [\boldsymbol{r}_t]_{\ell'}\}$ is zero for $\ell \neq \ell'$.

Now the problem reduces into investigate eigenvalues of $n$ 2-by-2 matrices. For any given $\ell = 1, 2, \ldots, n$, we have studied this problem in Section S-V.2 (type (1) and type (3) fixed points).

Specifically, the perfect recovery point $\boldsymbol{P} = \boldsymbol{I}$, $\boldsymbol{q} = \boldsymbol{r} = \boldsymbol{0}$ is stable if and only if $\lambda_{\max}(\boldsymbol{J}(\boldsymbol{P})) \leq 0$, where $\boldsymbol{J}(\boldsymbol{P})$ is defined in (S-65). Similar to the analysis of the type (3) fixed points in Section S-V.2, the condition that both eigenvalues of the matrix in (S-66) is non-positive implies

$$\tfrac{1}{2}([\boldsymbol{\Lambda}]_{\ell,\ell} - [\widetilde{\boldsymbol{\Lambda}}]_{\ell,\ell} + \alpha[\widetilde{\boldsymbol{\Lambda}}]_{\ell,\ell}) \leq \tau\overline{\eta^2} \tag{S-67}$$

$$\text{and} \quad \alpha(\tau\overline{\eta^2} - [\boldsymbol{\Lambda}]_{\ell,\ell}) \leq \tfrac{\tau\overline{\eta^2}}{[\widetilde{\boldsymbol{\Lambda}}]_{\ell,\ell}}(\tau\overline{\eta^2} - [\boldsymbol{\Lambda}]_{\ell,\ell} + [\widetilde{\boldsymbol{\Lambda}}]_{\ell,\ell}), \tag{S-68}$$

for all $\ell = 1, 2, \ldots, n$. The inequality (S-67) is the first inequality of (15) in Claim 1 in the main text.

Next, we investigate the condition when the trivial fixed point of the origin $\boldsymbol{P} = \boldsymbol{0}$ and $\boldsymbol{q} = \boldsymbol{r} = \boldsymbol{0}$ is unstable. Put $\boldsymbol{P} = \boldsymbol{0}$ into (S-66), we get a diagonal matrix

$$
\begin{bmatrix}
\tau([\boldsymbol{\Lambda}]_{\ell,\ell} - \tau\overline{\eta^2}) & 0 \\
0 & [\widetilde{\boldsymbol{\Lambda}}]_{\ell,\ell}(\widetilde{\tau} - \tau) - \tau^2\overline{\eta^2}
\end{bmatrix}.
$$

When any eigenvalue of the above matrices for $\ell = 1, 2, \ldots, n$ is positive, this trivial fixed point will be unstable. A sufficient condition is the first eigenvalues of all matrices are positive:

$$
\tau\overline{\eta^2} < [\boldsymbol{\Lambda}]_{\ell,\ell} \text{ for all } \ell = 1, 2, \ldots, n. \tag{S-69}
$$

The above inequality is the second inequality of (15) in the main text. In addition, (S-69) implies (S-68) hold as the left hand side of (S-68) is negative. Now, we prove that (15) is a sufficient condition that the perfect fixed point is stable and the trivial fixed point is unstable.

$\square$

We further note that (15) is not a necessary condition. There may be a region that (S-69) does not hold, but the origin is still unstable, and the perfect recovery point is stable. Such region is hard to characterize analytically, and numerically, we found the training algorithms always converge to other bad fixed points (e.g. mode collapsing state, or a state that $\boldsymbol{P}$ and $\boldsymbol{q}$ are still zero, but $\boldsymbol{r}$ is non-zero. The situation of the latter is similar to the noninfo-2 phase in the $d = 1$ case, which converges to the type (2) fixed point). Further study on those bad fixed points will be established in future works under a more general model.