[Reviews · NeurIPS 2019]

Reviewer 1



Overall the paper is quite well written and convincing. The claimed rigorous results are not proven at all in the paper but proofs (which are based on known techniques, in particular developed for ML problems by recent works including some by the authors) are provided in the SM. Still I think that it would have been nice to give hints on how the proof works at least at a high level, just to convince the reader not keen to read the SM, or as an introduction for the more mathematically inclined one. The results are strong, the studied model is relevant despite being simple enough for complete analysis. The results in terms of ODEs/PDEs/SDEs are a bit harsh and do not give much insight by themselves (the equations are quite complicated) but the authors then provide nice numerical experiments to back up the results, and meaningful interpretation of the observed learning regimes, which I find very nice and enlightening. The paper is of high quality. Still there are a number of typos that should be corrected. I recommend acceptance. Update: I have read the authors' rebuttal and taken it into account.

Reviewer 2



In this paper, the authors derive a mean-field theory for the analysis of GAN training when the learning rate is not necessarily small. They show that the training dynamics of a large network is described by a smaller set of parameters that scale as the number of hidden features. The dynamics of the order parameter obey a collection of ODEs while the microscopic parameters follow stochastic dynamics. The source of the noise in the dynamics is the noise in the data (here modeled as a spiked covariance matrix) and the generative model. They provide detailed analysis and simulation of a simple model with two hidden features and a simple quadratic loss function. Analysis of the temporal evolution of the order parameters reveals several dynamical phases. In one phase the weights of the generator converge to high overlap with that of the data model, signifying a success of feature retrieval by the generator. In another dynamical phase, the weights of the student oscillate around the correct values without ever converging. In the last phase, the system exhibits mode collapsing, where the generator finds only some of the features. They find that the bifurcating parameters between these solutions are the total amount of noise — the mean level of noise in the system, given by the mean of the noise in the data, and the noise level used for the generator. Interestingly, the macroscopic dynamics of the order parameters reveals an interesting hierarchical interaction when one feature is learned after the other. I think the authors should have focused more on this result, and perhaps compare it to known analysis of simple GAN (I do not know any, but either this is an exciting and new result or the authors should compare it with previous findings). However, I also have some criticism: - The authors introduce the macroscopic overlap matrix M, which aggregates all the order parameters into one matrix, but it makes it unclear. Different indices range of these matrix means different things, which make it hard to follow. - The result that the macroscopic dynamic is deterministic and accurate at the asymptotic regime with scaling 1/sqrt{N}, and that the microscopic dynamic is stochastic seem trivial to me. These are the starting point for any mean-field theory. While indeed these assumptions can, and often should be proven rigorously, the underlying microscopic noise, and the self-averaging of the teacher and student weights in the model is enough to support this claim. To conclude, I think this paper shows interesting findings on the training dynamics of GANs. I suggest that the authors focus more on the result they gain by analyzing the dynamics of the order parameters, and discuss further its meaning and applications. The further discussion can replace comparison between the deterministic macroscopic dynamics and the stochastic microscopic one — which seem trivial. ** update ** Despite the authors' response, I still find the use of the concatenated matrix M confusing. Nevertheless, it may be the better choice of due to the short format limitation. Either way, I still find this work interesting and believe it should be published. I am also satisfied with the authors' response to my comments about the emphasis on the stochastic microscopic dynamics. I agree that given previous works on a statistical analysis of GANs, which I am not well familiar with, it is a point important to make. I would suggest the authors add a couple of sentences explaining and emphasizing this, for the like of me that find their statements trivial.

Reviewer 3



The analyses performed in this study are concrete and sound good. It is an interesting direction to apply the theoretical framework of scaling limits of stochastic processes into the training of GANs. This work revealed that the training dynamics is highly nontrivial even in a simple setting with single-layer models (i.e., a linear model as the generator and simple perceptron as the discriminator). It enables us to theoretically quantify the effects of stochastic noise on the dynamics and, especially, to observe appropriately sized noises which lead to the success of feature recovery and prevent oscillation and model collapsing. My major concern is how one can generalize this analysis into more general settings. ・When d>1: Although the dynamics shown in Figure 1 is the case of d=2, the phase diagram shown in Supplementary Material corresponds to d=1. Since practical GANs usually suppose the multiple features (d>1), it seems to be better to show a detailed analysis of d=2. It would be helpful to enrich the description on d=2 which is briefly discussed in lines 45-50 in Supplementary Material and show its phase diagram. ・Norm constraints on U (in line 77): Authors say that we can set columns of U to be orthonormal with each other without loss of generality. Is there any preconditioning or whitening method which can normalize arbitrary U to a matrix with orthonormal columns? Assuming this orthonormal U will be essential in the analysis of example 1 because it should be impossible to find the solution V=U when U and V have different norms. It is better to describe how one can normalize U which determines an unknown data distribution. ・Case of nonlinear generators: The linear generator (1) seems to an ideal setting and one natural question is how one can generalize this analysis into non-linear generators. It will be helpful to remark on the case of non-linear generators and explain in what point the analytical evaluation become intractable in such non-linear generators as a mixture of Gaussians and VAE. Presentational issues: - line 1, “shallow GAN”: This word seems to be hardly used. Precisely speaking, this study investigated a solvable GAN with single-layer generative and discriminative models. - Supplementary Material, line 51, “heuristic derivations”: In what sense is it heuristic? The derivation shown here seems to be asymptotically exact when n is sufficiently large. Is there any approximation or do you mean that the derivation is not mathematically rigorous but intuitive? Typos: Caption in Fig 1: model collapsing -> mode collapsing? Line 8: this analysis provide”s” Line 297: our analysis provide”s” --- After rebuttal ------------------------------------------------------- Authors' response solved most of my concerns. I am still not so convinced of how the analysis can be extended into more general settings, but Authors did some additional works for understanding the phase diagram of d>=2 and promised to add comments on more complicated generators such as ReLU and learnable P_\tilde{c}. So, I keep my score and am looking forward to seeing the revised paper.

[Author Response · NeurIPS 2019]

We thank the reviewers for their useful comments, which help us improve the clarity and quality of our paper. We address these comments point by point. The paper will also be revised accordingly, with additional details given there.

***Reviewer 1** suggested us to provide a short "proof sketch".* We will present a brief summary of our proof strategy. It generally follows the classical recipe for establishing the weak convergence of interactive particle systems. Our proof sketch will also provide a roadmap to the SM, which contains all the technical details of the proof.

***Reviewer 1** asked why we choose $\log\cosh(x)$ as the regularization function $H(x)$ in example 1.* In fact, any convex function with its minimum reached at zero would be fine, for example $H(x) = |x|$ or $x^2$. The function $H(x) = \log\cos(x)$ is just a convenient special case since its derivative $H'(x) = \tanh(x)$ is smooth and bounded.

***Reviewer 1** asked what "a sufficiently strong regularizer" means in example 1.* It means to use a sufficiently large $\lambda$. In our experiments, $\lambda > 1$ is sufficient. In such cases, the regularized variables, e.g. $\boldsymbol{w}^\top \boldsymbol{w} - 1$ and each diagonal term of $\boldsymbol{V}^\top \boldsymbol{V} - \boldsymbol{I}$ in (4) are restricted to around 0 during the training process.

***Reviewer 1**'s additional comments:*
- We will emphasize that $P_{\tilde{c}}$ is fixed in our model. Meanwhile, a learnable $P_{\tilde{c}}$ is indeed an interesting idea to explore.
- By "analyzing the long-time behavior', we meant the investigation of the local stability of the ODE in Section 4. We will clarify this point.
- The matrix $\boldsymbol{M}(t)$ is $(2d+1) \times (2d+1)$, where $d$ is the number of features in the true data model (1).
- Thank you for catching the various typos and other miscellaneous issues. We will fix these and post the code.

***Reviewer 2** suggested avoiding using a single matrix M for the macroscopic states.* We agree with the reviewer that $\boldsymbol{M}$ is simply a concatenation of different macroscopic states $\boldsymbol{P}$, $\boldsymbol{q}$, $\boldsymbol{S}$, $\boldsymbol{r}$ and $z$, all of which have concrete meanings. However, we believe that there is indeed some benefit in introducing $\boldsymbol{M}$, which serves to streamline later presentations (e.g. those in Sec. 3.1). In the revised paper, we will start our discussions by first defining $\boldsymbol{P}$, $\boldsymbol{q}$, $\boldsymbol{S}$, $\boldsymbol{r}$ and $z$ as the actual macroscopic states we study. We will then introduce $\boldsymbol{M}$ and emphasize that it is just a convenient and compact notation.

***Reviewer 2** remarked that the notions of the macroscopic and microscopic states are standard in physics.* It is indeed a standard assumption in statistical physics that the macroscopic states of large systems tend to converge to deterministic values due to self-averaging. However, we note that the mean-field regime in our work was not considered in previous theoretical analysis of GAN. For example, a series of recent work [11-16] considers a different scaling regime where the learning rate goes to zero but the system dimension $n$ stays fixed. In that regime, the microscopic dynamics are deterministic even with the presence of the microscopic noise. In contrast, we study the regime where the learning rate is fixed but the dimension $n \to \infty$. This setting allows us to quantify the effect of training noise in the learning dynamics. Thus, we believe our work provides new and valuable insights into the theoretical understanding of GAN.

***Reviewer 2** suggested adding more discussions on the learning dynamics.* In the revised paper, we will use the one extra page available to us to provide more discussions on the hierarchical interactions of the macroscopic variables. Specifically, we will consider the special setting where the ratio of the generator's and the discriminator's learning rates $\tilde{\tau}/\tau \to 0$. This allows us to further simplify the ODE in (8) and to show that the learning process will be a combination of two dynamics with two different time scales. The discriminator's process is associated with the faster time scale, whereas the generator's process relates to the slower one.

***Reviewer 3** suggested us to extend the discussion of the phase diagram to $d \geq 2$ cases.* After the initial submission we have done some additional work on understanding the phase diagram for cases when $d \geq 2$. Accordingly, we will expand the current discussions on line 45–50 in the SM. Specifically, we will present local stability analysis about the phases of Info-1 and Noninfo-1 for $d = 2$. The analytical characterizations of the other phases are much more challenging when $d \geq 2$. Instead of showing analytical results, we will characterize these phases numerically by directly computing the eigenvalues of the Jacobian of the limiting ODEs.

***Reviewer 3** asked how one can orthonormalize the columns of a general $\boldsymbol{U}$.* Suppose $\boldsymbol{U}$ in the data model (1) is not orthogonal. We can always rewrite the product $\boldsymbol{U}\boldsymbol{c}$ as $(\boldsymbol{U}\boldsymbol{R})(\boldsymbol{R}^{-1}\boldsymbol{c})$, where $\boldsymbol{c}$ is the original random variable in the feature space, and $\boldsymbol{R} \in \mathbb{R}^{d \times d}$ is a matrix that orthogonalizes and normalizes the columns of $\boldsymbol{U}$. We can then study an equivalent system where the new feature vector is $\boldsymbol{R}^{-1}\boldsymbol{c}$.

***Reviewer 3** asked how one can generalize our analysis to handle non-linear generators.* One possible extension is to add a non-linear function, *e.g.* ReLU, to the linear data model (1) and the generator (2). Our analysis technique can be extended to handle this case, where one can still obtain a differential equation in the scaling limit. Another possible extension is a mixture of Gaussian model, where we need to treat $P_{\tilde{c}}$ as a learnable distribution rather than a fixed one. (This was also mentioned by Reviewer 1.) We will comment on these possible extensions in the revised paper.

***Reviewer 3** pointed out several typos and presentation issues.* Thank you. We will correct them in the revised version. By "heuristic derivation", we meant that certain steps presented in that section were not fully rigorous as we directly discard higher-order terms without any justification, in order to highlight the main ideas. In Section S-IV in the SM, we rigorously justify these steps by providing bounds on those terms.

[Meta-Review · NeurIPS 2019]

This paper proposes a model for adversarial learning of the spiked covariance model. The proposed model is solvable in the high-dimensional limit, and this paper provides analysis of the dynamics of the proposed model, both in the macroscopic and microscopic levels. The macroscopic dynamics is represented by a set of ODEs as shown in Theorem 1, whereas the microscopic dynamics is described stochastically, in terms of the SDE (10). Local stability analysis of the macroscopic dynamics is also provided in Section 4, which shows that a proper level of background noise in the learning process can prevent oscillation and help stabilize the learning in some cases. The review scores are above the acceptance threshold. I feel that the model studied in this paper is not really a GAN in the usual sense of this term, because the common goal of GAN is generative modeling, i.e., to learn the (possibly complex) distribution from which real data are generated, whereas the generator in the model in general may not have capability of implementing a true data distribution. Nevertheless, all the reviewers agree that this paper contains nice contributions. They are also basically satisfied with the authors' rebuttal. I would therefore like to recommend acceptance of this paper.